# Federated Learning over Connected Modes

**Dennis Grinwald**[1,2]**, Philipp Wiesner**[2]**, Shinichi Nakajima**[1,2,3]
[1]BIFOLD, [2]TU Berlin, [3]RIKEN Center for AIP
`{dennis.grinwald, wiesner, nakajima}@tu-berlin.de`

## Abstract

Statistical heterogeneity in federated learning poses two major challenges: slow global training due to conflicting gradient signals, and the need of personalization for local distributions. In this work, we tackle both challenges by leveraging recent advances in *linear mode connectivity* — identifying a linearly connected low-loss region in the parameter space of neural networks, which we call solution simplex. We propose federated learning over connected modes (FLOCO), where clients are assigned local subregions in this simplex based on their gradient signals, and together learn the shared global solution simplex. This allows personalization of the client models to fit their local distributions within the degrees of freedom in the solution simplex and homogenizes the update signals for the global simplex training. Our experiments show that FLOCO accelerates the global training process, and significantly improves the local accuracy with minimal computational overhead in cross-silo federated learning settings.

## 1 Introduction

Federated learning (FL) [1] is a decentralized machine learning paradigm that facilitates collaborative model training across distributed devices while preserving data privacy. However, in typical real applications, statistical heterogeneity—non-identically and independently distributed (non-IID) data distributions at clients—makes it difficult to train well-performing models. To tackle this difficulty, various methods have been proposed, e.g., personalized FL [2], clustered FL [3], advanced client selection strategies [4], robust aggregation [5], regularization strategies [6], and federated meta- and multi-task learning approaches [7, 8]. These methods aim either at training a global model that performs well on the global distribution [9], or, as it is common in personalized FL, at training multiple client-dependent models each of which performs well on its local distribution [10]. These two aims often pose a trade-off—a model that shows better local performance tends to suffer from worse global performance, and vice versa. In this work, we aim to develop a FL method that improves local performance compared to state-of-the art methods without sacrificing global performance.

Our approach leverages recent findings on *mode connectivity* [11–13]—the existence of low-loss paths in the parameter space between independently trained neural networks—and its applications [14]. These works show that minima for the same task are typically connected by simple low-loss curves, and that this connectivity benefits training for multi-task and continual learning. In particular, the authors show that embracing mode connectivity between models improves accuracy on each task and remedies the risk of catastrophic forgetting.

In this paper, we leverage such effects, and propose federated learning over connected modes (FLOCO), where the clients share and together train a *solution simplex*—a linearly connected low-loss region in the parameter space. Specifically, FLOCO represents clients as points within the standard simplex based on the similarity between their gradients, and assigns each client a specific subregion of the simplex. Clients then participate in FL by sampling different models within their assigned subregions and sending back the gradient information to update the vertices of the global solution simplex (see

38th Conference on Neural Information Processing Systems (NeurIPS 2024).

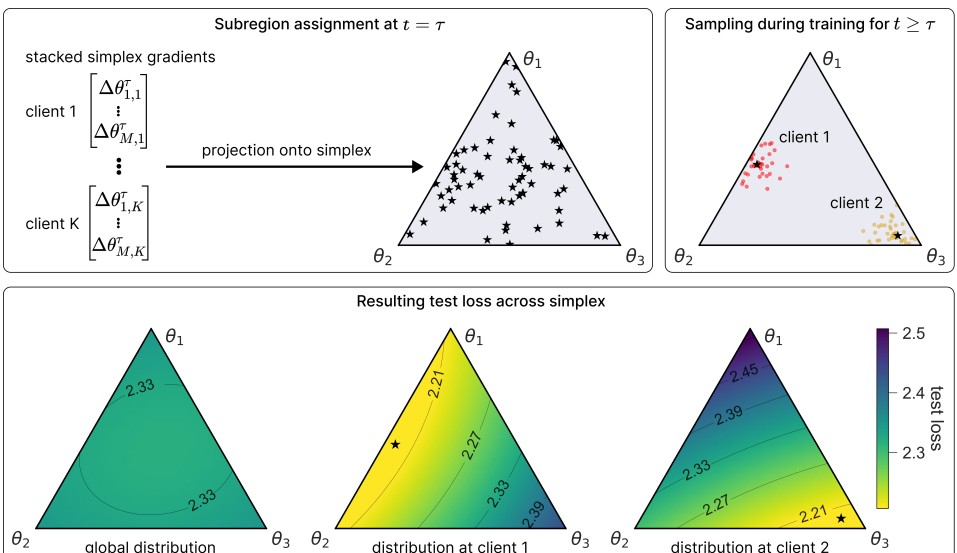

Figure 1: FLOCO expresses each client as a point (⋆ in the top-center plot) by projecting the gradient signals onto the simplex, so that similar clients are close to each other. In each communication round, each client uniformly samples points in the neighborhood of their projected point (top-right plot), and jointly train the solution simplex. The lower row shows the resulting test loss on the solution simplex, where the loss for the global distribution (left) is uniformly small, while the losses for individual local distributions (center for client 1 and right for client 2) are small around their projected points.

Fig.1). This method facilitates collaborative training through the common solution simplex, while allowing for client-specific personalization according to their local data distributions.

Our experiments show that FLOCO outperforms common FL baselines (FedAvg [1], FedProx [15]) and state-of-the-art personalized FL approaches (FedRoD [16], APFL [17], Ditto [18], FedPer [19]) on both local and global test metrics—without introducing significant computational overhead—in cross-silo FL settings. We also demonstrate additional benefits of FLOCO, including better uncertainty estimation, improved worst client performance, and smaller divergence of gradient signals.

Our main contributions are summarized as follows:

- We propose FLOCO, a novel FL method that trains a solution simplex for mitigating the statistical heterogeneity of clients, and demonstrate its state-of-the-art performance for local personalized FL.

- We propose a simple projection method to express clients as points in the standard simplex based on the gradient signals, and establish a procedure of subregion assignments.

- We conduct experimental evaluations on semi-artificial and real-world FL benchmarks with detailed analyses of the behavior of FLOCO, which give insights into how the mechanism improves performance compared to the baselines.

We provide implementations of FLOCO in the FL frameworks FL-bench [20] and Flower [21]. Our code is publicly available: `https://github.com/dennis-grinwald/floco`.

## 2  Background

In this section, we briefly explain the concepts behind federated learning and mode connectivity, which form the backbone of our approach. The symbols that we use throughout the paper are listed in Table 5 in Appendix.

## 2.1 Federated Learning

Assume a federated system where the server has a global model $g_0$ and the $K$ clients have their local models $\{g_k\}_{k=1}^K$. FL aims to obtain the best performing models $\{g_k^*\}_{k=0}^K$ such that

$$g_0^* = \operatorname{argmin}_{g_0} F^*(g_0) \equiv \sum_{k=1}^K p(k) F_k^*(g_0), \tag{1}$$

$$g_k^* = \operatorname{argmin}_{g_k} F_k^*(g_k) \quad \text{for } k = 1, \ldots, K, \tag{2}$$

$$\text{where } F_k^*(g) = \mathbb{E}_{(\boldsymbol{x},y) \sim p_k(\boldsymbol{x},y)} \left[ f(g, (\boldsymbol{x}, y)) \right].$$

Here, $p(k)$ is the normalized population of data samples for the $k$-th client, $p_k(\boldsymbol{x}, y)$ is the data distribution for the client $k$, and $f(g, (\boldsymbol{x}, y))$ is the loss, e.g., cross-entropy, of the model $g$ on a sample $(\boldsymbol{x}, y) \in \mathbb{R}^I \times \{1, \ldots, L\}$, where $I$ is the dimension of an input data sample. *Global* [22] and *personalized* [10] FL aim to approximate $g_0^*$ and $\{g_k^*\}_{k=1}^K$, respectively, by using the training data $\mathcal{D} = \{\mathcal{D}_k\}_{k=1}^K$ observed by the clients. Throughout the paper, we assume that all models are neural networks (NNs) $\widehat{y} = g_k(\boldsymbol{x}; \boldsymbol{w}_k)$ with the same architecture, and represent the model $g_k$ with its NN parameters $\boldsymbol{w}_k \in \mathbb{R}^D$, i.e., we hereafter represent $g_k(\boldsymbol{x}; \boldsymbol{w}_k)$ by $\boldsymbol{w}_k$ and thus denote, e.g., $F_k^*(g_k)$ by $F_k^*(\boldsymbol{w}_k)$. Let $N = \sum_{k=1}^K N_k$ be the total number of samples, where $N_k = |\mathcal{D}_k|$.

For the independent and identically distributed (IID) data setting, i.e., $p_k(x, y) = p(x, y), \forall k = 1, \ldots, K$, the global and personalized FL aim for the same goal, and the minimum loss solution for the given training data is

$$\widehat{\boldsymbol{w}}_0 = \widehat{\boldsymbol{w}}_k = \operatorname{argmin}_{\boldsymbol{w}} F(\boldsymbol{w}) \equiv \sum_{k=1}^K \frac{N_k}{N} F_k(\boldsymbol{w}), \tag{3}$$

$$\text{where } F_k(\boldsymbol{w}) = \frac{1}{N_k} \sum_{(\boldsymbol{x},y) \in \mathcal{D}_k} f(\boldsymbol{w}, (\boldsymbol{x}, y)).$$

In this setting, Federated Averaging (FedAvg) [1],

$$\boldsymbol{w}_0^{t+1} = \boldsymbol{w}_0^t + \sum_{k \in \mathcal{S}^t} \frac{N_k}{N} \cdot \Delta \boldsymbol{w}_k^{t+1} \text{ for } t = 1, \ldots, T, \tag{4}$$

is known to converge to $\widehat{\boldsymbol{w}}_0$, and thus solve Eq. (3). Here, $\mathcal{S}^t$ is the set of clients that participate the $t$-th communication round, and $\Delta \boldsymbol{w}_k^{t+1} = \boldsymbol{w}_k^{t+1} - \boldsymbol{w}_0^t$ is the update after $T'$ steps of the local gradient descent,

$$\breve{\boldsymbol{w}}^{t'+1} = \breve{\boldsymbol{w}}^{t'} - \gamma \boldsymbol{\nabla} F_k(\breve{\boldsymbol{w}}^{t'}), \text{ for } t' = 1, \ldots, T', \tag{5}$$

where $\breve{\boldsymbol{w}}^0 = \boldsymbol{w}_0^t$, $\breve{\boldsymbol{w}}^{T'} = \boldsymbol{w}_k^{t+1}$, and $\gamma$ is the step size. FedAvg has been further enhanced with, e.g., proximity regularization [23], auxiliary data [24], and ensembling [25].

On the other hand, in the more realistic non-IID setting, where $\boldsymbol{w}_0^* \neq \boldsymbol{w}_k^*$, FedAvg and its variants suffer from slow convergence and poor local performance [26]. To address such challenges, Ditto [18] was proposed for personalized FL, i.e., to approximate the best local models $\{\boldsymbol{w}_k^*\}_{k=1}^K$. Ditto has two training phases: it first trains the global model $\widehat{\boldsymbol{w}}_0$ by FedAvg, then trains the local models with proximity regularization to $\widehat{\boldsymbol{w}}_0$, i.e.,

$$\widehat{\boldsymbol{w}}_k = \operatorname{argmin}_{\boldsymbol{w}_k} \widetilde{F}_k(\boldsymbol{w}_k, \widehat{\boldsymbol{w}}_0) \equiv F_k(\boldsymbol{w}_k) + \frac{\lambda}{2} \|\boldsymbol{w}_k - \widehat{\boldsymbol{w}}_0\|_2^2,$$

where $\lambda$ controls the divergence from the global model. Ditto has been shown to outperform many other non-IID FL methods, including the client clustering method HYPCLUSTER, adaptive federated learning (APFL), which interpolates between a global and local models [27], Loopless Local SGD (L2SGD), which applies global and local model average regularization [28], and MOCHA [7], which fits task-specific models through a multi-task objective.

## 2.2 Mode Connectivity and Solution Simplex

Freeman and Bruna (2017) [29], as well as Garipov et al. (2018) [12], discovered the mode connectivity in the NN parameter space—the existence of simple regions with low training loss between two well-trained models from different initializations. Nagarajan and Kolter (2019) [13] showed that the path is linear when the models are trained from the same initialization, but with different ordering of training data. Frankle et al. (2020) [30] showed that the same pre-trained models stay linearly-connected after fine-tuning with gradient noise or different data ordering.

Benton et al. (2021) [31] found that the low loss connection is not necessarily in 1D, and [32] showed that a simplex,

$$\mathcal{W}(\{\boldsymbol{\theta}_m\}) = \left\{ \boldsymbol{w}_\alpha(\{\boldsymbol{\theta}_m\}) = \sum_{m=1}^{M+1} \alpha_m \boldsymbol{\theta}_m; \boldsymbol{\alpha} \in \Delta^M \right\}, \tag{6}$$

within which any point has a small loss, can be trained from randomly initialized endpoints.

Here, $\{\boldsymbol{\theta}_m \in \mathbb{R}^D\}_{m=1}^{M+1}$ are the endpoints or vertices of the simplex, and $\Delta^M = \{\boldsymbol{\alpha} \in [0,1]^{M+1}; \|\boldsymbol{\alpha}\|_1 = 1\}$ denotes the $M$-dimensional standard simplex. This simplex learning is performed by finding the endpoints that (approximately) minimize

$$\mathbb{E}_{(\boldsymbol{x},y) \sim p(\boldsymbol{x},y)} \left[ \mathbb{E}_{\boldsymbol{w} \sim \mathcal{U}_{\mathcal{W}(\{\boldsymbol{\theta}_m\})}} [f(\boldsymbol{w}, (\boldsymbol{x}, y))] \right], \tag{7}$$

where $\mathcal{U}_{\mathcal{W}}$ denotes the uniform distribution on a set $\mathcal{W}$. During training, one model realization $w_\alpha$ from the simplex gets sampled and its gradient update wrt. the loss, e.g. cross-entropy, gets backpropagated to the simplex endpoints $\{\boldsymbol{\theta}_m\}_{m=1}^{M+1}$.

## 3 Proposed Method

In this section, we introduce our approach, where the mode connectivity is leveraged for collaborative training between personalized client models.

### 3.1 Federated Learning over Connected Modes (FLOCO)

The main idea behind FLOCO is to assign subregions of the solution simplex (6) to clients in such a way that similar clients train neighboring (and overlapped) regions, while enforcing (linear) connectivity to all other client's subregions. The connectivity constraint systematically regularizes client training and allows for efficient collaboration between them.

The subregion assignments need to reflect the similarity between the clients. To this end, FLOCO expresses each client as a point in the standard simplex, based on the gradient update signals. Specifically, it applies the *Euclidean projection onto the positive simplex* [33] with the Riesz s-Energy regularization [34], which gives well spread projections that preserve the similarity between the client's gradient signals as much as possible. Once the clients are projected onto the standard simplex as $\{\boldsymbol{\alpha}_k \in \Delta^M\}_{k=1}^K$, we assign the L1-ball with radius $\rho$ around $\boldsymbol{\alpha}_k$, i.e., $\mathcal{R}_k = \{\boldsymbol{\alpha} \in \Delta^M; \|\boldsymbol{\alpha} - \boldsymbol{\alpha}_k\|_1 \leq \rho\}$, to the $k$-th client. Note that the gradient update signals are informative for the subregion assignment only after the (global) model is trained to some extent. Therefore, the subregion assignment is performed after $\tau$ FL rounds are performed. Before the assignment, i.e., $t \leq \tau$, all clients train the whole standard simplex $\mathcal{R}_k = \Delta^M, \forall k$, which corresponds to a simplex learning version of FedAvg.

Starting from randomly initialized simplex endpoints $\{\boldsymbol{\theta}_m\}_{m=1}^{M+1}$, FLOCO performs the following steps for each participating client $k \in \mathcal{S}^t$ in each communication round $t$:

1. The server sends the current endpoints $\{\boldsymbol{\theta}_m^t\}_{m=1}^{M+1}$ to the client $k$.
2. The client $k$ performs simplex learning only on the assigned subregion $\mathcal{R}_k$ as a local update.
3. The client sends the local update of the endpoints to the server.

This way, FLOCO is expected to learn the global solution simplex $\{\boldsymbol{w}_\alpha; \alpha \in \Delta^M\}$, while allowing personalization to local client distributions within the solution simplex. Algorithm 1 shows the main steps.

Although the simplex learning can be applied to all parameters, our preliminary experiment showed that applying simplex learning only of the parameters in the last fully-connected layer (while point-estimating the other parameters) is sufficient. Therefore, our FLOCO only applies the simplex learning to the last layer, which gives other benefits including applicability to fine-tuning of pre-trained models, and significant reduction of computational and communication costs, as shown in Section 4.3.

Below, we describe detailed procedures of client projection, local and global updates in the communication rounds, and inference in the test time.

**Algorithm 1:** Federated Learning over Connected Modes (FLOCO).

**Input** : number of communication rounds $T$, number of clients $K$, simplex dimension $M$, subregion assignment round $\tau$, subregion radius $\rho$

1  $\{\boldsymbol{\theta}_m^0\}_{m=1}^{M+1} \leftarrow$ **initialize_simplex**$(M)$

2  $\mathcal{R}_k \leftarrow \Delta^M, \forall k = 1, \ldots, K$  // set all client subregions to the whole standard simplex

3  **for** $t = 1$ **to** $T$ **do**

4      **if** $t = \tau$ **then**

5          $\{\{\Delta\boldsymbol{\theta}_{m,k}^\tau\}_{m=1}^{M+1}\}_{k=1}^K \leftarrow$ **collect_and_stack_gradients**()

6          $\{\boldsymbol{\alpha}_k\}_{k=1}^K \leftarrow$ **client_representation**$(\{\{\Delta\boldsymbol{\theta}_{m,k}^\tau\}_{m=1}^{M+1}\}_{k=1}^K)$

7          $\{\mathcal{R}_k\}_{k=1}^K \leftarrow$ **assign_subregions**$(\{\boldsymbol{\alpha}_k\}_{k=1}^K, \rho)$

8      $\mathcal{S}^t \leftarrow$ **choose_participating_clients**()

9      **for** $k \in \mathcal{S}^t$ **do**

10          $\{\boldsymbol{\theta}_{m,k}^{t+1}\}_{m=1}^{M+1} \leftarrow$ **local_update**$(\{\boldsymbol{\theta}_{m,k}^t\}_{m=1}^{M+1}, \mathcal{R}_k)$

11      $\{\boldsymbol{\theta}_m^{t+1}\}_{m=1}^{M+1} \leftarrow$ **global_update**$(\{\{\boldsymbol{\theta}_{m,k}^{t+1}\}_{m=1}^{M+1}\}_{k\in\mathcal{S}^t})$

## 3.2 Client Gradient Projection onto Standard Simplex

We explain how to obtain the representations $\{\boldsymbol{\alpha}_k \in \Delta^M\}$ of the clients in the standard simplex such that similar clients are located close to each other, while all clients are well-spread across the simplex.

At communication round $t = \tau$, FLOCO uses the gradient updates of the endpoints $\{\Delta\boldsymbol{\theta}_{m,k}^\tau\}_{m=1}^{M+1}$ as a representation of the client $k$. We concatenate the gradients for the $M + 1$ endpoints into a $((M + 1) \cdot D)$-dimensional vector, and apply the PCA projection onto the $M$ dimensional space, yielding $\boldsymbol{\kappa}_k \in \mathbb{R}^M$ as a low dimensional representation. To project $\{\boldsymbol{\kappa}_k\}$ onto the standard simplex $\Delta^M$, we solve the following minimization problem:

$$\min_{z>0} \quad \sum_{i,j} \frac{1}{\|\widehat{\boldsymbol{\beta}}_i(z) - \widehat{\boldsymbol{\beta}}_j(z)\|_2^2}, \tag{8}$$

$$\text{subject to:} \quad \widehat{\boldsymbol{\beta}}_k(z) = \operatorname{argmin}_{\frac{\boldsymbol{\beta}_k}{z} \in \Delta^{M-1}} \|\boldsymbol{\beta}_k - \boldsymbol{\kappa}_k\|_2^2. \tag{9}$$

The objective function in Eq. (8) is the Riesz s-Energy [34], a generalization of potential energy of multiple particles in a physical space, and therefore its minimizer correponds to the state where particles are well spread across the space. The minimization in the constraint (9) corresponds to the *Euclidean projection onto the positive simplex* [33], which forces $\{\boldsymbol{\beta}_k\}$ to keep the locations of the PCA projections $\{\boldsymbol{\kappa}_k\}$ of the clients. Fortunately, this minimization problem (for a fixed $z$) is convex, and can be efficiently solved (see Appendix A). We solve the main problem (8) by computing $\widehat{\boldsymbol{\beta}}_k(z)$ on a 1D grid in $z \in [0, 1]$ with the interval 0.001, and set the representations of the clients to $\boldsymbol{\alpha}_k = \frac{\widehat{\boldsymbol{\beta}}_k(\widehat{z})}{\widehat{z}}$, where $\widehat{z}$ is the minimizer of Eq. (8).

## 3.3 Communication Round: Local and Global Updates

In the $t$-th communication round, the server sends the current endpoints $\{\boldsymbol{\theta}_m^t\}_{m=1}^{M+1}$ to the participating clients $\mathcal{S}^t$. Then, each client $k \in \mathcal{S}^t$ draws one sample per mini-batch from the uniform distribution $\mathcal{A} = \{\boldsymbol{\alpha}_b\}_{b=1}^B \sim \mathcal{U}_{\mathcal{R}_k}$ on the assigned subregion and applies $T'$ local updates,

$$\breve{\boldsymbol{\theta}}_m^{t'+1} = \breve{\boldsymbol{\theta}}_m^{t'} - \alpha_m \cdot \gamma \cdot \boldsymbol{\nabla} F_k(\boldsymbol{w}_{\boldsymbol{\alpha}}), \tag{10}$$

to the endpoints with $\boldsymbol{\alpha}$ sequentially chosen from $\mathcal{A}$.[1] Here $\breve{\boldsymbol{\theta}}_m^0 = \boldsymbol{\theta}_m^t, \breve{\boldsymbol{\theta}}_m^{T'} = \boldsymbol{\theta}_{m,k}^{t+1}$. The local updates $\{\Delta\boldsymbol{\theta}_{m,k}^{t+1} = \boldsymbol{\theta}_{m,k}^{t+1} - \boldsymbol{\theta}_m^t\}_{m=1}^{M+1}$ are sent back to the server, which updates the endpoints as

$$\boldsymbol{\theta}_m^{t+1} = \boldsymbol{\theta}_m^t + \sum_{k\in\mathcal{S}^t} \frac{N_k}{N} \cdot \Delta\boldsymbol{\theta}_{m,k}^{t+1}. \tag{11}$$

---

[1]Note, that we do not rely on any regularizer that forces the diversity of the endpoints, as in [32]. In FLOCO, the diversity of local client distributions prevents the simplex endpoints from collapsing to a single point.

As explained in Section 3.1, the client subregions are initially set to the whole simplex $\Delta^M$ before the subregion assignment is performed at $t = \tau$, which corresponds to a straightforward application of the simplex learning to FedAvg. After the subregion assignment, FLOCO uses the degrees of freedom within the solution simplex to personalize clients models.

## 3.4  FLOCO$^+$

We can further enhance the personalized FL performance of FLOCO by additionally fine-tuning a local model as in Ditto [18]. In this extension, called FLOCO$^+$, each client personalizes the global endpoints $\{\widehat{\boldsymbol{\theta}}_m^0 = \boldsymbol{\theta}_m\}_{m=1}^M$ by local gradient descent to minimize the Ditto objective, i.e.,

$$\{\widehat{\boldsymbol{\theta}}_m^k\} = \mathrm{argmin}_{\{\boldsymbol{\theta}_m\}} \widetilde{F}_k(\{\boldsymbol{\theta}_m\}, \{\widehat{\boldsymbol{\theta}}_m^0\})$$

$$\equiv \mathbb{E}_{\boldsymbol{\alpha} \sim \mathcal{U}_{\mathcal{R}_{z_k}}} \left[ F_k(\boldsymbol{w}_\alpha(\{\boldsymbol{\theta}_m\})) \right] + \tfrac{\lambda}{2} \sum_{m=1}^{M+1} \|\boldsymbol{\theta}_m - \widehat{\boldsymbol{\theta}}_m^0\|_2^2.$$

## 3.5  Inference

With the trained endpoints $\{\widehat{\boldsymbol{\theta}}_m = \boldsymbol{\theta}_m^T\}_{m=1}^{M+1}$, we simply use $\boldsymbol{w}_{\widehat{\boldsymbol{\alpha}}_0}(\{\widehat{\boldsymbol{\theta}}_m\}_{m=1}^{M+1})$ as the global model, where $\widehat{\boldsymbol{\alpha}}_0 = \frac{1}{M+1}\mathbf{1}_{M+1}$ with $\mathbf{1}_D$ denoting the $D$-dimensional all one vector. For local models, we use $\{\boldsymbol{w}_{\widehat{\boldsymbol{\alpha}}_k}(\{\widehat{\boldsymbol{\theta}}_m\}_{m=1}^{M+1})\}_{k=1}^K$ where $\widehat{\boldsymbol{\alpha}}_k = \boldsymbol{\alpha}_k$. For FLOCO$^+$, we fine-tune the corresponding subspace regions $\mathcal{R}_{z_k}$ for $E$ local epochs.

# 4  Experiments

In this section, we experimentally show the advantages of FLOCO and FLOCO$^+$ over the baselines.

## 4.1  Experimental Setting

**Datasets and models.**  To evaluate our method, we perform image classification on the CIFAR-10 [35] and FEMNIST [36] datasets. For CIFAR-10, we train a CNN (CifarCNN) from scratch, following [37], and fine-tune a ResNet-18 [38] pre-trained on ImageNet [39], as in [40]. For FEMNIST, we train a CNN (FemnistCNN) from scratch, as in [1], and fine-tune a SqueezeNet [41] pre-trained on ImageNet, following [40]. We provide a table with the training hyperparameters that we use for each dataset/model setting in Appendix B.

**Data heterogeneity for non-FL benchmarks.**  The FEMNIST dataset is an FL benchmark based on real data, where client heterogeneity is inherently embedded in the dataset. For CIFAR-10, we simulate statistical heterogeneity by two partitioning procedures. The first procedure by [42] partitions clients in equally sized groups and assigns each group a set of primary classes. Every client gets $q\,\%$ of its data from its group's primary classes and $(100 - q)\,\%$ from the remaining classes. We apply this method with $q = 80$ for five groups and refer to this split as *5-Fold*. For example, in CIFAR-10 *5-Fold*, 20 % of the clients get assigned 80 % samples from classes 1-2 and 20 % from classes 3-10. The second procedure, inspired by [43] and [44], draws the multinomial parameters of the client distributions $p_k(y) = \mathrm{Multi}(y; \boldsymbol{\phi}_k)$ from Dirichlet, i.e., $\boldsymbol{\phi}_k \sim \mathrm{Dir}_L(\beta)$, where $\beta$ is the concentration parameter controlling the sparsity and heterogeneity—$\beta \to \infty$ concentrates the mass to the uniform distribution (and thus homogeneous), while small $0 < \beta < 1$ generates sparse and heterogeneous non-IID client distributions.

**Baseline methods.**  Besides FedAvg [1] and FedProx [23] for global FL, we chose FedRoD [16], APFL [17], Ditto [18], and FedPer [19] as state-of-the-art personalized FL baselines.

**FLOCO Hyperparameters.**  For CifarCNN on the simulated non-IID splits Dir(0.3)/Five-Fold, we set $\tau = 250, M = 20/10, \rho = 0.1$. For FemnistCNN on FEMNIST we set $\tau = 250, M = 10, \rho = 0.5$. For pre-trained ResNet-18 on the simulated non-IID splits Dir(0.3)/Five-Fold we set $\tau = 50, M = 20/10, \rho = 0.1$ and for the pre-trained SqueezeNet on FEMNIST we set $\tau = 250, M = 3, \rho = 0.5$. We found those settings work well in our preliminary experiments, and conducted ablation study with other parameter settings in Appendix D. For the baselines, we follow the recommended parameter settings by the authors, which are detailed in Appendix B.

Table 1: Average global and local test accuracy.

| | CIFAR-10 | | | | | | | | FEMNIST | | | |
| | CifarCNN | | | | pre-trained ResNet-18 | | | | FemnistCNN | | pre-trained | |
| | 5-Fold | | Dir(0.3) | | 5-Fold | | Dir(0.3) | | | | SqueezeNet | |
|---|---|---|---|---|---|---|---|---|---|---|---|---|
| FedAvg | 60.36 | 60.38 | 60.74 | 60.78 | 75.33 | 76.94 | 68.59 | 59.27 | 78.83 | 79.84 | 75.13 | 75.51 |
| FedProx | 60.68 | 60.36 | 60.40 | 60.27 | 76.93 | 77.46 | 62.27 | 60.26 | 78.84 | 80.15 | 75.47 | 75.99 |
| FedPer | 40.23 | 65.42 | 33.90 | 67.86 | 68.64 | 84.06 | 50.84 | 85.05 | 50.76 | 73.83 | 64.03 | 74.43 |
| APFL | 60.56 | 60.33 | 60.55 | 60.65 | 53.25 | 46.46 | 50.97 | 44.57 | 4.95 | 4.98 | 38.21 | 58.86 |
| Ditto | 60.36 | 72.22 | 60.74 | 73.90 | 75.33 | 69.18 | 68.59 | 76.23 | 78.83 | 82.02 | 57.89 | 65.06 |
| FedRoD | 56.36 | 74.03 | 46.12 | 76.42 | 17.46 | 31.82 | 10.27 | 33.85 | 4.95 | 4.99 | 4.95 | 4.95 |
| FLOCO | **62.93** | 71.78 | **62.57** | 71.04 | **77.15** | 85.90 | **73.62** | 80.38 | **78.99** | 84.09 | **75.86** | 77.00 |
| FLOCO$^+$ | **62.93** | 75.08 | **62.57** | 76.50 | **77.15** | 84.88 | **73.62** | 85.89 | **78.99** | 84.75 | **75.86** | 82.41 |

Table 2: Average global and local expected test calibration error.

| | CIFAR-10 | | | | | | | | FEMNIST | | | |
| | CifarCNN | | | | pre-trained ResNet-18 | | | | FemnistCNN | | pre-trained | |
| | 5-Fold | | Dir(0.3) | | 5-Fold | | Dir(0.3) | | | | SqueezeNet | |
|---|---|---|---|---|---|---|---|---|---|---|---|---|
| FedAvg | 24.08 | 25.61 | 22.95 | 24.51 | 13.77 | 19.57 | 13.48 | 19.57 | 12.40 | 16.86 | 15.54 | 20.43 |
| FedProx | 23.76 | 25.56 | 23.19 | 24.89 | 12.40 | 12.41 | 15.16 | 19.83 | 12.41 | 16.93 | 15.48 | 20.04 |
| FedPer | 47.75 | 28.22 | 56.39 | 25.70 | 19.73 | 11.19 | 38.48 | **10.88** | 38.44 | 21.68 | 28.28 | 22.31 |
| APFL | 23.30 | 25.01 | 22.19 | 23.91 | 28.39 | 33.39 | 20.02 | 26.01 | 4.95 | 4.98 | **7.6** | 15.82 |
| Ditto | 24.08 | 19.13 | 22.95 | 17.64 | 13.77 | 16.43 | 13.48 | 14.50 | 12.40 | 14.65 | 15.54 | 18.06 |
| FedRoD | 29.78 | 18.40 | 41.91 | 17.45 | 75.59 | 64.07 | 89.31 | 64.07 | 4.95 | 4.99 | 4.99 | 4.99 |
| FLOCO | **21.82** | 18.44 | **20.06** | 18.75 | **11.48** | **9.44** | **10.30** | 11.28 | **10.28** | 13.94 | 14.65 | 19.15 |
| FLOCO$^+$ | **21.82** | 17.69 | **20.06** | 16.50 | **11.48** | 12.42 | **10.30** | 11.98 | **10.28** | 13.87 | 14.65 | 15.35 |

**Evaluation criteria.** For the performance evaluation, we adopt two metrics, the test accuracy measured after the last communication round (ACC) and the time-to-best-accuracy (TTA), each for evaluating the global and local FL performance. ACC is the last test accuracy over $T$ communication rounds, i.e, $\text{ACC}(T) = \frac{1}{N_{\text{test}}} \sum_{i=1}^{N_{\text{test}}} \mathbb{1}(y_i = \arg\max g(\boldsymbol{x}_i; \widehat{\boldsymbol{w}}^T))$, where $\mathbb{1}(\cdot)$ is the indicator function that equals to 1 if the event is true and 0 otherwise. TTA evaluates the number of communication rounds needed to achieve the best baseline (FedAvg and Ditto in this paper) test accuracy, i.e., $\text{ACC}_{\text{FedAvg}}(T)$. We report TTA improvement, i.e. the TTA of the baseline, e.g. FedAvg, divided by the TTA of the benchmarked method, e.g. FLOCO. Moreover, we report the expected-calibration-error (ECE) [45], a common measure that evaluates the quality of uncertainty estimation of a trained model, for the last communication round.

## 4.2 Results

Table 1 and 2 summarize the main experimental results, where FLOCO and FLOCO$^+$ consistently outperform the baselines across the different experiments in terms of global (red) and local (blue) test accuracy, as well as test ECE. The global and local test metrics are measured after the last communication round and averaged over 5 different seed runs. The best performances are highlighted in bold, while the underlined entries indicate the settings that did not converge properly. Note that the global test performances of FEDAVG and DITTO, as well as FLOCO and FLOCO$^+$, are the same since they use the same global model. Below we report on detailed observations.

**Global and local FL test accuracy.** We first evaluate the global and local test performance on CIFAR-10 with the non-IID data splits generated by the 5-Fold and $\text{Dir}(\beta)$ procedures, as well as the natural non-IID data splits in the FEMNIST dataset. Table 1 shows the test accuracies on CIFAR-10 with CifarCNN trained from random initialization (left) and ResNet-18 fine-tuned from the ImageNet pre-trained model (center), respectively. It also shows the test accuracies on FEMNIST with FemnistCNN trained from random initialization (left) and SqueezeNet fine-tuned from the ImageNet pre-trained model (right). We clearly see that FLOCO and FLOCO$^+$ outperform all baselines in terms of average local (blue) test accuracy by up to $6\%$, as well as global (red) by up to $5\%$.

**Calibration.** We evaluate and benchmark the quality of uncertainty estimation of all methods. For this purpose we evaluate the global as well as average local ECE on each model-dataset combination for each baseline on the test dataset and show the results in Table 2. As shown, FLOCO and FLOCO$^+$ achieve better Expected Calibration Error (ECE) across all settings, with two exceptions: training a pre-trained ResNet-18 on the CIFAR-10 Dir(0.3) split and a pre-trained SqueezeNetV1 on FEMNIST. In the first case, the average local ECE for FLOCO and FLOCO$^+$ is slightly worse than that of FedPer, suggesting mild overconfident for some clients. In the second case, the next best method (APFL) yields a significantly lower global test accuracy than our method, making a fair comparison of their ECE difficult.

**Worst client performance.** We evaluate the average local and global test accuracies of the worst 5% of clients, a standard approach for assessing potential biases of the FL method toward specific clients or client groups [46]. The worst 5% client performance on all CIFAR-10/model combinations is evaluated over 5 trial runs, with results shown in the table on the right. We observe that FLOCO achieves the highest performance among worst-performing clients across all settings, with a 17% improvement over FedAvg, and up to 1.5% over the next best baseline.

Table 3: Average *local* test accuracy for the 5% worst performing clients on CIFAR-10.

| | CIFAR-10 (CifarCNN) | |
|---|---|---|
| | 5-Fold | Dir(0.3) |
| FedAvg | *44.0 ± 0.02* | *42.93 ± 0.03* |
| FedProx | *43.87 ± 0.02* | *43.23 ± 0.03* |
| FedPer | *52.67 ± 0.02* | *51.01 ± 0.02* |
| APFL | *43.27 ± 0.02* | *46.36 ± 0.03* |
| Ditto | *58.20 ± 0.03* | *58.69 ± 0.03* |
| FedRoD | *60.20 ± 0.02* | *61.12 ± 0.03* |
| FLOCO$^+$ | ***61.73 ± 0.02*** | *61.13 ± 0.03* |

**Time-to-accuracy.** Similar to Table 1, we plot the TTA improvement for FLOCO. In particular, we show the TTA improvement of FLOCO over FedAvg and FedProx, and the TTA improvement of FLOCO$^+$ over Ditto, FedPer and FedRod, as all these methods include local fine-tuning. We report all TTAs in Table 4. The underlined entries indicate the cases where the test accuracies of our methods exceed the baseline method's maximum accuracy already at the initial evaluation round, while the entries labeled '*x1.0*' represent the instances where our methods take the same evaluation rounds to achieve the baseline method's maximum accuracy, i.e., comparable in terms of TTA. In addition to test accuracy, we also observe an improvement in Time-to-Accuracy (TTA) for our method across all settings.

Table 4: Improvements for global and *local* time-to-accuracy.

| | CIFAR-10 | | | | | | | | FEMNIST | | | |
|---|---|---|---|---|---|---|---|---|---|---|---|---|
| | CifarCNN | | | | pre-trained ResNet-18 | | | | FemnistCNN | | pre-trained | |
| | 5-Fold | | Dir(0.3) | | 5-Fold | | Dir(0.3) | | | | SqueezeNet | |
| FLOCO vs. FedAvg | x5.5 | *x4.6* | x3.4 | *x3.1* | x1.3 | *x1.8* | x1.2 | *x8.0* | x1.7 | *x1.2* | x1.1 | *x1.1* |
| FLOCO vs. FedProx | x5.1 | *x4.9* | x3.3 | *x3.8* | x1.0 | *x1.8* | x1.2 | *x9.0* | x3.0 | *x1.2* | x1.0 | *x1.1* |
| FLOCO$^+$ vs. Ditto | x5.5 | *x2.3* | x3.4 | *x2.1* | x1.3 | *x2.0* | x1.2 | *x1.7* | x1.7 | *x4.0* | x9.0 | *x4.0* |
| FLOCO$^+$ vs. FedPer | x1.0 | *x1.5* | x1.0 | *x1.3* | x1.6 | *x1.5* | x1.5 | *x1.5* | x7 | *x7* | x7.0 | *x2.7* |
| FLOCO$^+$ vs. FedRoD | x9.4 | *x1.6* | x24.5 | *x1.3* | x10 | *x10* | x10 | *x10* | x7 | *x7* | x10 | *x10* |

## 4.3 Analysis and Discussion

In this section, we provide further analyses and discussion on FLOCO.

**Solution structure in simplex.** First, we confirm that FLOCO uses the degrees of freedom within the solution simplex for personalization. To this end, we draw approximately 500 uniformly distributed points in the solution simplex, and evaluate the global and the local test accuracy of the corresponding models. Figure 1 (bottom row) shows the global test accuracy (left most) and the local test accuracy (center and right) for two clients. As expected, for the global test dataset the solution simplex performs uniformly well across all its area, while the losses for the two individual local client distributions are small around their projected points ($\star$). This result indicates that the heterogeneous sharing of the solution simplex across the clients properly works as designed.

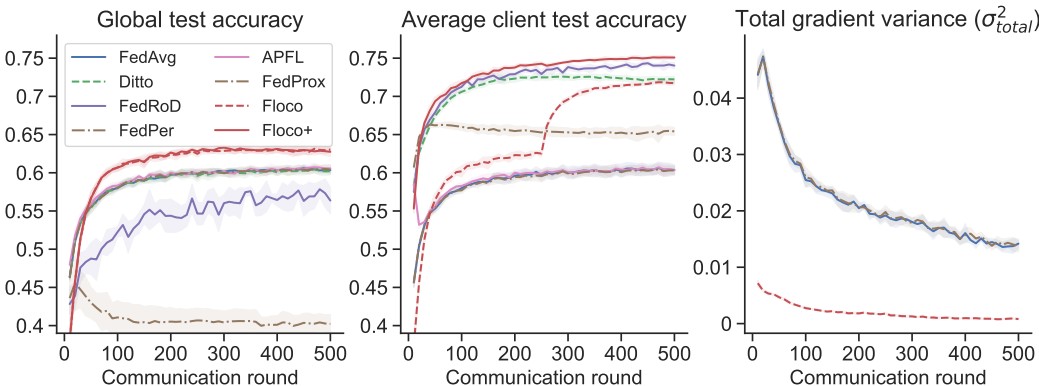

Figure 2: Global (left) and average local (center) test accuracy for CifarCNN on CIFAR-10, 5-Fold. For FLOCO, we can clearly observe a jump in average local test accuracy at $\tau = 250$, which is a result of our subregion assignment. Right shows the total variance of the gradients for the last fully-connected layer.

**Gradient variance reduction and stability of training.** Figure 2 shows the test accuracy curves during training for global (left) and average local (center) test accuracies of different methods with the standard deviation over 5 trials as shadows. We observe that FLOCO and FLOCO$^+$ not only converge faster than the global and pFL baselines respectively, but also show small standard deviation across trials. The latter implies that our systematic regularization through the solution simplex stabilizes the training dynamics significantly. Figure 2 (right) shows the total gradient variance—the sum of the variances of the updates $\Delta \boldsymbol{w}_k^t = \boldsymbol{w}_k^t - \boldsymbol{w}_0^{t-1}$ for FedAvg and FedProx (which almost overlap with each other), and $\Delta \boldsymbol{\theta}_{m,k}^t = \boldsymbol{\theta}_{m,k}^t - \boldsymbol{\theta}_{m,0}^{t-1}$ for FLOCO, respectively. More specifically, we compute the variance over the last fully-connected layer, given by

$$\sigma_{\text{total}}^2(t) = \sum_{k \in \mathcal{S}^t} \|\Delta \boldsymbol{w}_k^t - \tfrac{1}{|\mathcal{S}^t|} \sum_{k \in \mathcal{S}^t} \Delta \boldsymbol{w}_k^t\|_2^2 \tag{12}$$

for FedAvg and FedProx, and by

$$\sigma_{\text{total}}^2(t) = \tfrac{1}{M+1} \sum_{m=1}^{M+1} \sum_{k \in \mathcal{S}^t} \|\Delta \boldsymbol{\theta}_{m,k}^t - \tfrac{1}{|\mathcal{S}^t|} \sum_{k \in \mathcal{S}^t} \Delta \boldsymbol{\theta}_{m,k}^t\|_2^2. \tag{13}$$

We have not plotted the gradient variances of FLOCO$^+$ and the other pFL methods, since those are the same as for FLOCO and FEDAVG, respectively. As discussed in [47, 48], a small total variance indicates effective collaborations with consistent gradient signals between the clients, leading to better performance. From the figure, we see that the total gradient variance of FLOCO is much lower and more stable, in terms of standard deviation, than the baseline methods, which, together with its good performance observed in Table 1, is consistent with their discussion. The variance reduction with FLOCO implies that the degrees of freedom of the solution simplex can absorb the heterogeneity of clients to some extent, making the gradient signals more homogeneous. Moreover, [49] argued that the last classification layer has the biggest impact on performance, implying that reducing the total variance of the classification layer, as FLOCO does with simplex learning, is most effective. As we show in the Appendix C, applying simplex learning to only the last layer, instead of learning a simplex in the whole parameter space, achieves faster personalized and global convergence.

**Computational complexity.** If the batch size is one, simplex training adds $O(\pi \cdot M)$ computational complexity for each layer, where $\pi$ is the parameter complexity of the layer, e.g., $\pi = d \cdot L$ for a fully connected layer with $d$ input and $L$ output neurons, and $M$ is the simplex dimension [32]. For FLOCO, this additional complexity only applies to the classification layer. For inference, no additional complexity arises, compared to FedAvg, because inference is performed by the single model corresponding to the cluster center. Since the most modern architectures, e.g., ResNet-18 and Vision Transformer (ViT) [50], have parameter complexity of $O(\mathbb{G}_{\text{FE}}) \gg O(\mathbb{G}_{\text{C}})$, where $\mathbb{G}_{\text{FE}}$ and $\mathbb{G}_{\text{C}}$ are the complexities of the feature extractor and the classification layer, respectively, the additional training complexity, applied only to the classification layer, of FLOCO is ignorable, i.e., $O(\mathbb{G}_{\text{FE}}) \gg O(\mathbb{G}_{\text{C}} \cdot M)$. The same applies to the communication costs: since the simplex learning is applied only to the classification layer, the increase of communication costs are ignorable compared to the communication costs for the feature extractor.

# 5 Related Work

There are few existing works that apply simplex learning to federated learning. [37] proposed SuPerFed, which enforces a low loss simplex between independently initialized global and client models, yielding good personalized FL performance. This approach builds on [27], which finds optimal interpolation coefficients between a global and local model to improve personalized FL. However, their simplex is restricted to be 1D, i.e., a line segment, and the global model performance is comparable to the plain FedAvg. Moreover, they train a solution simplex over all layers between global and local models, which is computationally expensive and limits its applicability to training *from scratch*. This should be avoided if pre-trained models are available [40, 51]. Our method generalizes to training low-loss simplices of higher dimensions in a FL setting, tackles both the global and personalized FL objectives, is applicable to pre-trained models, and shows significant performance gains by employing our proposed subregion assignment procedure. In Table 7 of Appendix E we benchmark FLOCO against the SuPerFed baseline on the CIFAR-10, 5-Fold, as well as Dir(0.5) splits using both a CifarCNN trained from scratch as well as a pre-trained ResNet18 on both global as well as local test performance, where we observe that FLOCO outperforms SuPerFed both in terms of global as well as local accuracy in all settings.

# 6 Limitations

In this work, we only evaluate our method on cross-silo FL settings with up to 100 clients. Unlike cross-device FL, which typically involves a much larger set of stateless clients (i.e., clients with limited data that hinders reliable modeling), our approach assumes stateful clients, each with sufficient data to enable effective grouping of similar clients. While our current analysis focuses on cross-silo FL, extending our method to the cross-device setting is an important direction for future research. Additionally, a thorough theoretical analysis of our approach remains a future research objective.

# 7 Conclusion

FL on highly non-IID client data distributions remains a challenging problem and a very actively researched topic. Recent works tackle non-IID FL settings either through global or personalized FL. While the former aims to find a single optimal set of parameters that fit a global objective, the latter tries to optimize multiple local models each of which fits the local distribution well. These two different objectives may pose a trade-off, that is, personalized FL might adapt models to strongly to local distributions which might harm the global performance, while global FL solutions might fit none of the local distributions if the local distributions are diverse. In this paper, we addressed this issue by leveraging the mode-connectivity of neural networks. Specifically, we propose FLOCO, where each client trains an assigned subregion within the solution simplex, which allows for personalization, and at the same, contributes to learning a well-performing global model. FLOCO achieves state-of-the-art performance in both global and personalized FL, with minimal computational and communication overhead during training and no overhead during inference.

Promising future research directions include better understanding the decision-making process of solution simplex training through global and local explainable AI methods [52–54]. Furthermore, we want to apply our approach to continual learning problems and FL scenarios with highly varying client availability [55, 56].

# Acknowledgements

This work was funded by the German Ministry for Education and Research as BIFOLD - Berlin Institute for the Foundations of Learning and Data (ref. BIFOLD24B).

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

## Appendix

This appendix provides a nomenclature, details to our optimization problem and experimental setup, as well as additional results and insights.

Table 5: Nomenclature.

| Symbol | Description |
|---|---|
| $k = 1, \ldots, K$ | Clients |
| $t = 1, \ldots, T$ | Communication rounds |
| $t' = 1, \ldots, T'$ | Local training iterations |
| $\mathcal{S}^t$ | Participating clients in round t |
| $B$ | Mini-batch size |
| $\gamma$ | Client gradient descent step size |
| $\mathcal{D}_k$ | Training data of client $k$ |
| $N$ | Total number of samples |
| $N_k$ | Number of samples at client $k$ |
| $p_k(\boldsymbol{x}, y)$ | data distribution of client $k$ |
| $\boldsymbol{w}_0^t \in \mathbb{R}^D$ | Global model at round $t$ |
| $\boldsymbol{w}_k^t \in \mathbb{R}^D$ | Model of client k at round $t$ |
| $\Delta^M = \{\boldsymbol{\alpha} \in [0,1]^{M+1}; \|\boldsymbol{\alpha}\|_1 = 1\}$ | $M$-dimensional standard simplex |
| $\boldsymbol{\theta}_1^t, \ldots, \boldsymbol{\theta}_{M+1}^t$ | Simplex endpoints at round $t$ |
| $\boldsymbol{w}_\alpha = \sum_{m=1}^{M} \alpha_m \boldsymbol{\theta}_m$ | Model parameters at a point $\boldsymbol{\alpha} \in \Delta^M$ |
| $\rho$ | Subregion radius |
| $\mathcal{R}_k$ | Assigned subregion of client k |
| $\tau \in [1, \ldots, T]$ | Subregion assignment round |
| $\boldsymbol{\kappa}_k \in \mathbb{R}^M$ | Low dimensional representation of stacked gradient update $\{\Delta\boldsymbol{\theta}_{m,k}^\tau\}_{m=1}^{M+1}$ of client k |

## A    Optimization Problem

The Lagrangian of the lower-level optimization problem in (9) has the following formulation $\mathcal{L}(\boldsymbol{\alpha}_k, \lambda) = \frac{1}{2}\|\boldsymbol{\alpha}_k - \boldsymbol{\kappa}_k\|_2^2 + \lambda(\mathbf{1}^T\boldsymbol{\alpha}_k - z)$ with $\lambda \in \mathbb{R}$ being the Langrange multiplier. The Lagrangian can be further rewritten to $\mathcal{L}(\boldsymbol{\alpha}_k, \lambda) = \frac{1}{2}\|\boldsymbol{\alpha}_k - (\boldsymbol{\kappa}_k - \lambda\mathbf{1})\|_2^2 + \lambda(\mathbf{1}^T\boldsymbol{\kappa}_k - z) - \lambda^2 n$ such that the optimization problem reduces to solving

$$\min_{z \in \mathbb{R}} \quad \frac{1}{2}\|\boldsymbol{\alpha}_k - (\boldsymbol{\kappa}_k - \lambda\mathbf{1})\|_2^2 \tag{14}$$

$$\text{subject to:} \quad \boldsymbol{\alpha}_k \succeq \mathbf{0}. \tag{15}$$

The optimal solution of (14) is given by $\boldsymbol{\alpha}_k^* = [\boldsymbol{\kappa}_k - \lambda^*\mathbf{1}]_+$. Plugging it back into the Lagrangian we get the following dual function

$$\mathcal{L}(\boldsymbol{\alpha}_k, \lambda) = \frac{1}{2}\|[\boldsymbol{\kappa}_k - \lambda^*\mathbf{1}]_+ - (\boldsymbol{\kappa}_k - \lambda\mathbf{1})\|_2^2 + \lambda(\mathbf{1}^T\boldsymbol{\kappa}_k - z) - \lambda^2 n \tag{16}$$

$$= \frac{1}{2}\|[\boldsymbol{\kappa}_k - \lambda^*\mathbf{1}]_-\|_2^2 + \lambda(\mathbf{1}^T\boldsymbol{\kappa}_k - z) - \lambda^2 n. \tag{17}$$

Finding $\boldsymbol{\alpha}_k^*$ can be achieved by maximizing (17) using for example the bisection algorithm [57]. After that the projected points are obtained as $\boldsymbol{\alpha}_k^* = [\boldsymbol{\kappa}_k - \lambda^*\mathbf{1}]_+$.

## B    Training Hyperparameters

Table 6 summarizes all hyperparameters that were used for each dataset/model combination. We train CifarCNN on CIFAR-10 for a total of 500 communication rounds, ResNet-18 on CIFAR-10 for 100 communication rounds, FemnistCNN on FEMNIST for 350 rounds, and SqueezeNetV1 on FEMNIST for 1000 rounds. Moreover, we train each setting using a total of 100 clients, and for FEMNIST we select a randomly chosen subset of 100 total clients for each trial, of which we select 10 randomly to participate in training in each communication round, except for CifarCNN on CIFAR-10 where we select 30 out of 100 clients to participate in each round. We evaluate all clients after every ten communication rounds. For CIFAR-10 we train a CifarCNN with batch size 50 using SGD with a learning rate of 0.02, momentum of 0.5, and weight decay of $10^{-5}$, and a pre-trained

ResNet-18 with learning rate of batch size 32, using SGD with a learning rate of 0.01, momentum of 0.9, and weight decay of $10^{-4}$. For FEMNIST we train a pre-trained SqueezeNet with batch size 32 using SGD with a learning rate of 0.005, momentum of 0, weight decay of $10^{-4}$, and a FemnistCNN with batch size 32, learning rate 0.1, momentum of 0, weight decay of 0. For FedProx we set the proximity hyperparameter to $\mu = 0.01$ for all settings. For DITTO, FEDROD and FEDPER we set the local epochs to the same value as epochs for the global model, i.e. $E_{\text{DITTO}} = E$. All training hyperparameters for CIFAR-10 and FEMNIST on a FemnistCNN were taken from [37], CIFAR-10 on a pre-trained ResNet-18 from [51] and FEMNIST on pre-trained SqueezeNet from [40].

Table 6: Summary of used hyperparameters for training.

| Dataset/Model | T | K | $|S^t|$ | $e$ | $E/E_{\text{DITTO}}$ | $\gamma$ | mom. | wd | $\mu$ |
|---|---|---|---|---|---|---|---|---|---|
| CIFAR-10/CifarCNN | 500 | 100 | 30 | 50 | 5 | 0.02 | 0.5 | $10^{-5}$ | 0.01 |
| CIFAR-10/ResNet-18 | 100 | 100 | 10 | 32 | 5 | 0.01 | 0.9 | $10^{-4}$ | 0.01 |
| FEMNIST/FemnistCNN | 350 | 100 | 10 | 32 | 5 | 0.1 | 0.0 | 0.0 | 0.01 |
| FEMNIST/SqueezeNetV1 | 1000 | 100 | 10 | 32 | 5 | 0.005 | 0.0 | $10^{-4}$ | 0.01 |

## C   Simplex Learning on all NN parameters

In Figure 5, we compare the global (left) and average client (right) test accuracy of FLOCO and FLOCO-All, where the latter applies simplex learning to all NN parameters. As expected, FLOCO-All converges to the same global and average local test accuracy, but needs more communication rounds to do so, since it needs to train more parameters.

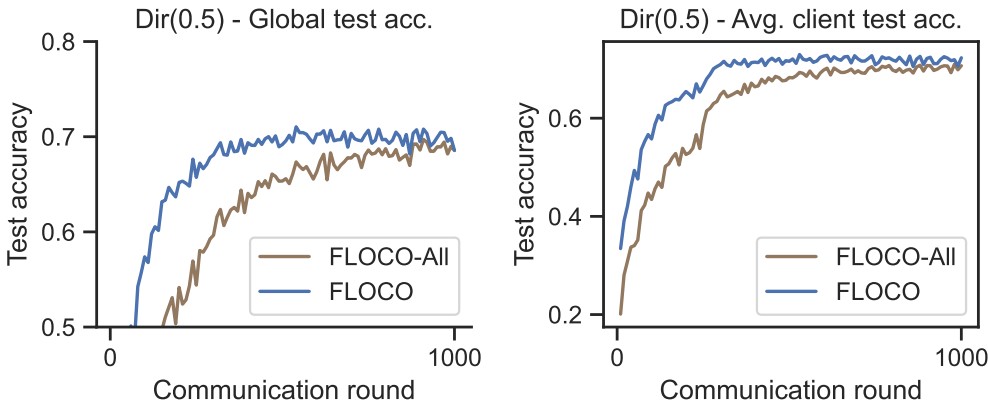

Figure 3: Global test accuracy.          Figure 4: Average local client test accuracy.

Figure 5: Comparing simplex learning on all network layers vs. only on the last fully-connected layer.

## D   Sensitivity to Parameter Setting

We investigate how stable the performance of FLOCO is for different hyperparameter settings. Specifically, we tested FLOCO with the combination of $\tau = 50, 100, 200$ (subregion assignment time step) and $\rho = 0.1, 0.2, 0.4$ (radius of subregions), and show the average local client and global test accuracy for CifarCNN on CIFAR-10 5-Fold in Figure 6. We observe that the average local client test accuracy (left) increases for earlier subregion assignment starting points $\tau$ and lower client subregion radiuses $\rho$, with the best reached test accuracy being approximately $4\%$ better than the worst, i.e., $82.79\%$ against $78.18\%$. The intuition for this is that earlier client specialization in less overlapping regions allows for better personalization. On the other hand, as can be observed in the right heatmap of Figure 6 the global test performance is less sensitive to the choice of these hyperparameters,

Table 7: Average global and *local* test accuracy on CIFAR-10.

| | CIFAR-10 | | | | | | | |
| --- | --- | --- | --- | --- | --- | --- | --- | --- |
| | CifarCNN | | | | pre-trained ResNet-18 | | | |
| | 5-Fold | | Dir(0.3) | | 5-Fold | | Dir(0.3) | |
| SuPerFed | 63.22 | *76.65* | 63.00 | *71.73* | 64.88 | *52.78* | 76.04 | *60.91* |
| FLOCO | **68.26** | ***80.92*** | **69.79** | ***74.64*** | **74.61** | ***87.38*** | **79.11** | ***82.29*** |

i.e., 70.66% against 69.30%. This is because, even after subregion assignment, the entire solution simplex remains to be trained, making the midpoint (global model) of the simplex less sensitive to the specialization process for client distributions.

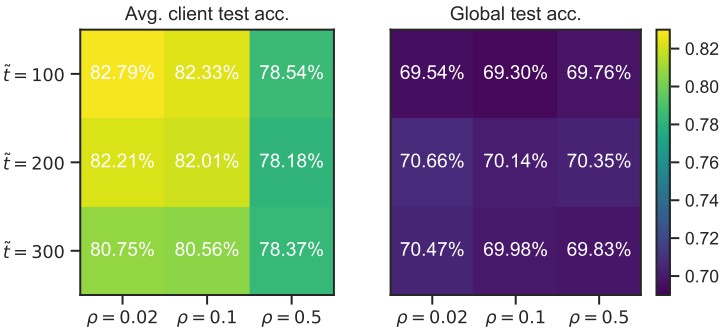

Figure 6: Local average client (left) and global (right) test accuracies for different subregion assignment time step $\tau$ and subregion radius $\rho$ settings.

# E  Comparing FLOCO to SuPerFed

We benchmark FLOCO against the SuPerFed baseline on the CIFAR-10, 5-Fold, as well as Dir(0.5) splits using both a CifarCNN trained from scratch as well as a pre-trained ResNet18, on both global as well as local test performance. As shown in Table 7, FLOCO outperforms SuPerFed in all settings. Note, that for this benchmark we have implemented FLOCO as well as SUPERFED in the FL framework Flower [21].

