# OpenReview forum: "Federated Learning over Connected Modes"
_NeurIPS.cc/2024/Conference — NeurIPS 2024 poster_

### Official Review · Reviewer_a16f · 2024-07-07

**Soundness:** 3
**Presentation:** 3
**Contribution:** 3
**Rating:** 5
**Confidence:** 4

**Summary:**

This paper proposes a method for FL to train a simplex of linearly connected solutions with uniformly low loss. Clients are expressed in the simplex by projecting the gradient onto the simplex and similar clients are close to each other. In each communication round each client samples points in the neighborhood of their projected point and jointly train the solution simplex.

Each client is expressed as a point in the simplex based on the gradient update signals. The subregion assignments preserve the similarity of the clients by applying the Euclidean projection onto the positive simples with Riesz s-Energy regularization - this spreads the projections while maintaining the similarity between the clients. As the gradient update signals are informative for the subregion assignment only after the model has been trained for some time, the first stage of the algorithm is FedAvg with simplex learning.

**Strengths:**

The paper is well written and the proposed approach is novel and interesting. The method applies simplex learning to tackle personalized federated learning.

**Weaknesses:**

Some details of the algorithm should be better explained. Some baselines seem to be missing.

**Questions:**

- Could you clarify how simplex learning works in practice? Do you compute the loss and the gradients on each of the endpoints or do you sample one point in the simplex and backpropagate to update the endpoints? On which weights do you compute the loss? I think this is unclear from the paper and it would be beneficial to add a section “Simplex Learning” for the reader.
- From my understanding, the idea of using a shared simplex and then dividing it into subregions is to have both global and local classifiers and to incentivize knowledge transfer and robustness. I would also see two other baselines/ablations that the authors might want to try to corroborate that this is actually beneficial. First, fedavg simplex learning (no subregions, check global accuracy) and then local fine-tuning for personalization (compute local accuracy) personalization can be done with local simplex learning or standard learning. I guess this could be a way to study the impact of simplex learning by itself and the importance of subregions.
- As the authors mentioned in the related work, there are other personalized FL methods that apply simplex learning. Why did the authors not report their results as baseline? E.g. references 29 and 19.
- I’m not sure I fully buy the claims and observations in the paragraph on variance reduction (lines 255-272). The main issue with heterogeneity is convergence speed, so to claim some effect on this the authors should show faster convergence on the global accuracy. There is evidence that the effect of the subregions impacts local accuracy, but this is expected as there is better specialization - as it would be with a local fine-tuning stage.
- I suggest the authors also have a look at other more recent methods for global learning such as scaffold, feddyn, mime that could help with heterogeneity. They could also be used in conjunction with simplex learning.
- I found it odd that the authors decided to show the plots in Figure 2 using the simpleCNN and FEMNIST while from table 1 the cifar10 experiments show more edge of FLOCO over the baselines.
- Basically, the method only applies to the last FC layer - did the authors try it on the full model?

**Limitations:**

Limitations are not clearly expanded and should be discussed in a section of the paper.

---

> ### Author Rebuttal · Authors · 2024-08-07
>
> We are very pleased and thankful for Reviewer a16f for their thorough review. In particular we are thankful that the reviewer found our method novel and interesting, and the soundness, presentation, contribution and experimental setup of our paper good.
> In the following, we will answer the reviewer’s remarks and questions in more detail:
>
> ***Reviewer’s remark:*** “Some details of the algorithm should be better explained. Some baselines seem to be missing.”
> ***Author’s answer:*** We will explain the algorithm in more detail in the camera ready version. Which parts should be explained better in particular?  We report on new comparisons with 5 other global AND local FL baselines.  As shown in Figures 1 and 2 in the rebuttal pdf and Tables 1-4 in the global response, our FLOCO method outperforms all the tested baselines (the results were averaged over 5 different random seeds).  We implemented our FLOCO in the FL Bench [1] framework, and compared it with most of the baselines except SuPerFed, another state-of-the-art FL method that is not supported by FL Bench.  SuPerFed is implemented in our Flower-based framework, and compared with our original implementation of FLOCO.  Note that FLOCO run on the FL Bench framework and that run on our Flower-based framework do not exactly match because of the mismatch in the aggregation order of clients.  Due to the time limitation, we could have run experiments in the FL base framework only on CIFAR10 Dir(0.5) on a SimpleCNN.   For the camera-ready we will run these comparisons on all dataset split-model combinations.
>
> ***Reviewer’s question 1:*** Could you clarify how simplex learning works in practice? Do you compute the loss and the gradients on each of the endpoints or do you sample one point in the simplex and backpropagate to update the endpoints? On which weights do you compute the loss? I think this is unclear from the paper and it would be beneficial to add a section “Simplex Learning” for the reader.
> ***Author’s answer:*** We sample one point in the simplex and backpropagate to update the endpoints.  The loss to be minimized in the simplex learning (where alpha is drawn from the uniform distribution) is given in Eq.(7) in the original submission.  Since FLOCO applies simplex learning only to the last layer, the weights in the other layers are point-estimated, which amounts to tying all end-points to a single point.  We will extend Section 2.2 with these details.
>
> ***Reviewer’s question 2:*** From my understanding, the idea of using a shared simplex and then dividing it into subregions is to have both global and local classifiers and to incentivize knowledge transfer and robustness. I would also see two other baselines/ablations that the authors might want to try to corroborate that this is actually beneficial. First, FedAvg simplex learning (no subregions, check global accuracy) and then local fine-tuning for personalization (compute local accuracy) personalization can be done with local simplex learning or standard learning. I guess this could be a way to study the impact of simplex learning by itself and the importance of subregions.
> ***Author’s answer:*** Thank you for your suggestion.  We show these results in Figs. 3 and 4 in the rebuttal pdf, and will show them in the camera ready.
>
> ***Reviewer’s question 3:*** As the authors mentioned in the related work, there are other personalized FL methods that apply simplex learning. Why did the authors not report their results as baseline? E.g. references 29 and 19.
> ***Author’s answer:*** The other simplex learning approach for federated learning, i.e. SuPerFed[1], is now included in Table 2 in the global response, which shows that FLOCO outperforms SuPerFed.  Moreover, we have benchmarked FLOCO against many other baselines for the SimpleCNN - Dit(0.5) setting and the results are shown in Table 1, where our method outperforms the other methods. For the camera-ready version, we will show complete comparisons on all dataset-model combinations
>
> ***Reviewer’s question 4:*** I’m not sure I fully buy the claims and observations in the paragraph on variance reduction (lines 255-272). The main issue with heterogeneity is convergence speed, so to claim some effect on this the authors should show faster convergence on the global accuracy. There is evidence that the effect of the subregions impacts local accuracy, but this is expected as there is better specialization - as it would be with a local fine-tuning stage.
> ***Author’s answer:*** FLOCO has M times more parameters to be trained because it learns the end-points of the solution simplex.  For this reason, the simplex learning is slower than the baselines until the 200th round, where FLOCO projects the clients onto the solution simplex and thus the variance of gradients gets significantly reduced.  After the 200th round, the consistent gradient signals of FLOCO accelerate the training and thus FLOCO converges to the model with higher global test accuracy.  We can observe  in Fig.2 (left) in the original submission steeper slopes in the global test accuracy after the 200th round, compared to FedAve and DITTO, although the difference is not as drastic as in the local accuracy.
>
> ***Reviewer’s question 5:*** I suggest the authors also have a look at other more recent methods for global learning such as scaffold, feddyn, mime that could help with heterogeneity. They could also be used in conjunction with simplex learning.
> ***Author’s answer:*** We have been focusing on comparison with many state-of-the-art baselines in the rebuttal, and unfortunately did not have time to evaluate the suggested baselines.
>
> ***Reviewer’s question 6:*** I found it odd that the authors decided to show the plots in Figure 2 using the simpleCNN and FEMNIST while from table 1 the cifar10 experiments show more edge of FLOCO over the baselines.
> ***Author’s answer:*** We chose simmpleCNN-FEMNIST because it shows a typical behavior.

---

> > ### Author Response · Authors · 2024-08-07
> > **Extension**
> >
> > ***Reviewer’s question 7:*** Basically, the method only applies to the last FC layer - did the authors try it on the full model?
> > ***Author’s answer:*** We show the result with FLOCO-all in Figures 5 and 6 in the rebuttal pdf, where we observe that applying FLOCO to all layers significantly reduces training speed.

---

> ### Comment · Reviewer_a16f · 2024-08-08
>
> Thank you for your rebuttal. I appreciate your effort and will take into account all your responses in my final score which I want to emphasize is positive.
> - In Q2 I've suggested more than one option for personalization. Could you clarify how you implemented the experiments for this ablation? I don't understand if the simplex baseline is implying some personalization (local fine-tuning) or not.
> - Regarding Q4, I understand what you mean by variance reduction now, but in the context of FL and optimization, this has a specific meaning (see SCAFFOLD).  By checking again table 1 I see the improvement over simple baselines such as fedavg also in global accuracy. However, since the baselines for global models are not the most recent ones (besides the newly tested feddyn, which is generally broken)  I suggest the authors tone down their claims in the contributions regarding "FLOCO outperforms state-of-the-art approaches in both global and local". It is however interesting the effect of their method on global accuracy hence reducing the effect of heterogeneity.
> - One last comment is about scalability. Could you comment on the possibility of making your method work in more realistic settings with thousands of clients (e.g. inaturalist, gldv2) ? What are the challenges you foresee? this could be expanded in the limitations section.

---

> ### Author Response · Authors · 2024-08-13
>
> Dear Reviewer a16f,
>
> Thank you very much for your additional comments.  Below we give temporal answers, and we will report on additional experiments with the suggested baselines by the discussion deadline.
>
> Best,
> Authors
>
>
> ***Reviewer’s question:*** In Q2 I've suggested more than one option for personalization. Could you clarify how you implemented the experiments for this ablation? I don't understand if the simplex baseline implies some personalization (local fine-tuning).
>
> ***Author’s answer:*** We realized that we misunderstood your suggestions when submitting our rebuttal.  The curves labeled as FedAvg-Simplex Learning in Figs. 3 and 4 in the rebuttal pdf are the global (Fig.3) and the average local (Fig.4) test accuracy of simplex learning without client projection and WITHOUT personalization.  We apologize for our confusion. We are now evaluating the following methods and will report on the average local test accuracy by the discussion deadline:
>
> - Baseline 1: We run FedAvg with simplex learning, and then, apply the plain DITTO personalization to the midpoint of the simplex solution. Namely, no local simplex is used.
> - Baseline 2: We run FedAvg with simplex learning, and then, apply the DITTO personalization with local simplex learning.
> - FLOCO+: We run FLOCO (with client projection after 100 communication rounds), and apply the DITTO personalization with local simplex learning.
>
> Unfortunately, the cluster server in my group is under maintenance after the Neurips rebuttal phase ended, and we are running the additional experiments on local computers.  To make the experiments feasible, we have to down scale the experiment.  Namely, our simplex learning is with a (M=2)-simplex (instead of M=6 in the submitted paper) for a SimpleCNN on CIFAR10 with a Dirichlet(0.5) split.  To make a fair comparison with our FLOCO+, we will also report on FLOCO+ results with M=2-simplex.  We estimate that all experiments will finish by tomorrow (Tuesday) night.
>
>
> ***Reviewer’s question:*** Regarding Q4, I understand what you mean by variance reduction now, but in the context of FL and optimization, this has a specific meaning (see SCAFFOLD). By checking again table 1 I see the improvement over simple baselines such as fedavg also in global accuracy. However, since the baselines for global models are not the most recent ones (besides the newly tested feddyn, which is generally broken) I suggest the authors tone down their claims in the contributions regarding "FLOCO outperforms state-of-the-art approaches in both global and local". It is however interesting the effect of their method on global accuracy hence reducing the effect of heterogeneity.
>
> ***Author’s answer:*** We agree that our original claim of outperforming state-of-the-art global FL approaches is not supported by our experiments.  Following the reviewer’s suggestion, we will tone down our claim.
>
>
> ***Reviewer’s question:*** One last comment is about scalability. Could you comment on the possibility of making your method work in more realistic settings with thousands of clients (e.g. inaturalist, gldv2) ? What are the challenges you foresee? this could be expanded in the limitations section.
>
> ***Author’s answer:*** We assume that by scalability the reviewer means the situation where many clients do not observe sufficient numbers of  data samples.  If we would have thousands of clients, all of which observe sufficiently many data samples, we do not foresee critical problems because the projection matrix can be computed with subsamples, and the other computations are linear operations.  However, if we assume the generic FL setting, we cannot reasonably assign subregions to the low data clients, because the server does not get sufficient information about those clients.  Following the suggestions by Reviewer a16f, as well as Reviewer 44Bh, we will revise the paper so that our target is not the generic FL setting, but the cross-silo FL setting, where each client observes sufficiently many data samples. We appreciate the reviewer’s constructive suggestions.

---

> > ### Comment · Reviewer_a16f · 2024-08-14
> >
> > Thank you for your answer. I’ll wait for the experiments. As I mentioned in my review I think these are important to better understand how the method works:
> >
> > > From my understanding, the idea of using a shared simplex and then dividing it into subregions is to have both global and local classifiers and to incentivize knowledge transfer and robustness

---

> > > ### Author Response · Authors · 2024-08-14
> > >
> > > Here we report on the average local test accuracy of Baseline 1, Baseline 2, and FLOCO+. Additionally, we tested a new version of our method, FLOCO++, inspired by our observation that Baseline 1 was better than Baseline 2.
> > >
> > >
> > > - ***FLOCO++:*** We run FLOCO (with client projection after 100 communication rounds), and apply the DITTO personalization to the projected point of each client.
> > >
> > >
> > > As we mentioned in the previous comments, we used a (M=2)-simplex for a SimpleCNN on CIFAR10 with a Dirichlet(0.5) split for computational reasons (our cluster server is down until the weekend).  Below we report on the average local test accuracy at 100, 200, 300, 400, and 500 communication rounds.
> > >
> > >
> > > - ***Baseline 1*** (Global simplex learning + plain DITTO personalization to the midpoint):
> > > 68.92, 74.09, 76.05, 76.92, 77.02
> > >
> > >
> > > - ***Baseline 2*** (Global simplex learning + DITTO personalization with local simplex):
> > > 69.43, 74.08, 76.10, 76.96, 76.86
> > >
> > >
> > > - ***FLOCO+:***
> > > 69.60, 74.20, 76.10, 77.00, ***77.17***
> > >
> > >
> > > - ***FLOCO++:***
> > > 70.01, 74.32, 76.11, 77.08, ***77.41***
> > >
> > >
> > > As expected, the personalization boosts the local test accuracy in general, and the gain by FLOCO is reduced, compared to what we showed in the rebuttal pdf as Fig. 4. However, we still observe some gains by FLOCO’s projection strategy. We expect that the comparison with M=6 would show larger gains by our methods, since a larger space inside the simplex should better capture the characteristics of the clients.  We will conduct the experiment with M=6 next week when our cluster servers will work again, and use the results in the paper.  We will also investigate why the local simplex DITTO personalization (Baseline 2 and FLOCO+) is worse than the plain DITTO personalization (Baseline 1 and FLOCO++). Thank you for your patience in waiting for our experimental results until the last minutes, and thank you again for your valuable comments to improve the paper. We will add FLOCO++ as another version of our proposal in the paper.

---

> > > > ### Comment · Reviewer_a16f · 2024-08-14
> > > >
> > > > Thank you - I will take some time during the reviewers discussion to take into account these results. I think I don’t see the baseline with standard fedavg and personalization, am I correct ?

---

> > > > > ### Author Response · Authors · 2024-08-14
> > > > > **DITTO**
> > > > >
> > > > > FedAve + DITTO personalization corresponds to the baseline "DITTO" in the original submission, of which the results are reported in Fig. 2 and Tables 1-3.

---

> > > > > > ### Comment · Reviewer_a16f · 2024-08-14
> > > > > >
> > > > > > Thank you - this helps.

---

### Official Review · Reviewer_5sib · 2024-07-11

**Soundness:** 3
**Presentation:** 3
**Contribution:** 3
**Rating:** 6
**Confidence:** 4

**Summary:**

In this study, the authors tackle challenges in federated learning by introducing FLOCO, which uses linear mode connectivity to identify a solution simplex in neural network weight space. This approach allows for personalized client model training within the simplex, while also enabling efficient updates to both global and local models.

**Strengths:**

The paper introduces an interesting application of the linear mode connectivity approach in a personalized FL (pFL) setting. This method can be considered an alternative to distance-based and similarity-based pFL algorithms. The algorithm is evaluated under various experimental setups and compared with standard and personalized FL algorithms.

**Weaknesses:**

To strengthen their empirical findings, the authors could benefit from leveraging the bilevel optimization literature for analyzing the convergence of their algorithm.

**Questions:**

$\textbf{Q1}.$  Choosing an appropriate experimental setup is often overlooked in the pFL setting. Note that the following two scenarios are inappropriate for pFL experiments: $\textbf{(1)}$ Clients have enough data to train their local models independently, eliminating the need to participate in an FL system. $\textbf{(2)}$ The data distribution of the clients is identical or nearly identical. This typically results in high accuracy for FedAvg, making it challenging to justify using pFL methods. It would be informative to evaluate your method in experimental setups where local training and FedAvg fail to provide good results and compare your method's performance with SOTA  pFL methods, especially methods with decoupled parameter space such as FedPer [1] and FedRep [2] and similarity-based methods such as pFedSim [3].

$\textbf{Q2}.$ As previously mentioned, evaluating the performance of local models on their respective local datasets is crucial in personalized federated learning (pFL). Including the performance metrics of these models (average performance when trained solely on local training data) would provide valuable insights into the experimental results.

[1] Arivazhagan, Manoj Ghuhan, et al. "Federated learning with personalization layers." arXiv preprint arXiv:1912.00818 (2019).

[2] Collins, Liam, et al. "Exploiting shared representations for personalized federated learning." International conference on machine learning. PMLR, 2021.

[3] Chen, Yizhu, et al. "PFedSim: An Efficient Federated Control Method for Clustered Training." IEEE Journal of Radio Frequency Identification 6 (2022): 779-782.

**Limitations:**

The discussion of the limitations of the proposed method is framed within the context of future directions. It is suggested to address these limitations separately, focusing on theoretical analysis and validation using real-world federated learning datasets suited for personalized FL (pFL).

---

> ### Author Rebuttal · Authors · 2024-08-07
>
> We are very pleased and thankful for Reviewer 5sib for their thorough review. In particular we are thankful that the reviewer found our application of linear mode connectivity to the FL setting interesting and the soundness, presentation, contribution and experimental setup of our paper good.
> In the following, we will answer the reviewer’s remarks and questions in detail:
>
> ***Reviewer’s remark:*** “To strengthen their empirical findings, the authors could benefit from leveraging the bilevel optimization literature for analyzing the convergence of their algorithm.”
> ***Author’s answer:*** Thank you for your suggestion.  We will consider theoretical analysis for future work.
>
> ***Reviewer’s question 1:*** “Choosing an appropriate experimental setup is often overlooked in the pFL setting. Note that the following two scenarios are inappropriate for pFL experiments: (1) Clients have enough data to train their local models independently, eliminating the need to participate in an FL system. (2) The data distribution of the clients is identical or nearly identical. This typically results in high accuracy for FedAvg, making it challenging to justify using pFL methods. It would be informative to evaluate your method in experimental setups where local training and FedAvg fail to provide good results and compare your method's performance with SOTA pFL methods, especially methods with decoupled parameter space such as FedPer [1] and FedRep [2] and similarity-based methods such as pFedSim [3].”
> ***Author’s answer:*** We agree that the originally chosen baselines are not sufficient, and thus we report on new comparisons with 5 other global AND local FL baselines.  As shown in Figures 1 and 2 in the rebuttal pdf and Tables 1-4 in the global response, our FLOCO method outperforms all the tested baselines (the results were averaged over 5 different random seeds).  We implemented our FLOCO in the FL Bench [1] framework, and compared it with most of the baselines except SuPerFed, another state-of-the-art FL method that is not supported by FL Bench.  SuPerFed is implemented in our Flower-based framework, and compared with our original implementation of FLOCO.  Note that FLOCO run on the FL Bench framework and that run on our Flower-based framework do not exactly match because of the mismatch in the aggregation order of clients.  Due to the time limitation, we could have run experiments in the FL base framework only on CIFAR10 Dir(0.5) on a SimpleCNN.   For the camera-ready we will run these comparisons on all dataset split-model combinations.
>
> ***Reviewer’s question 2:*** As previously mentioned, evaluating the performance of local models on their respective local datasets is crucial in personalized federated learning (pFL). Including the performance metrics of these models (average performance when trained solely on local training data) would provide valuable insights into the experimental results.
> ***Author’s answer:*** Since we have been focusing on additional comparisons with many state-of-the-art baselines, we could not have made it for comparison with this basic baseline.  We expect that the local training is at least much slower than the federated learning for the clients that only get a small number of samples.  We will conduct this basic experiment in the camera ready.
>
> ***Reviewer’s limitation remark:*** The discussion of the limitations of the proposed method is framed within the context of future directions. It is suggested to address these limitations separately, focusing on theoretical analysis and validation using real-world federated learning datasets suited for personalized FL (pFL).
> ***Author’s answer:*** We will follow your suggestion.
>
> [1] FL Bench: (https://github.com/KarhouTam/FL-bench)

---

> > ### Comment · Reviewer_5sib · 2024-08-12
> >
> > Thank you for your rebuttal. I would like to maintain my current score.

---

> ### Author Response · Authors · 2024-08-13
>
> Dear Reviewer bjGi,
>
> Thank you again for your constructive comments.  We will do our best to improve the paper, according to the reviewer’s comments.
>
> Best,
> Authors

---

### Official Review · Reviewer_44Bh · 2024-07-13

**Soundness:** 2
**Presentation:** 3
**Contribution:** 2
**Rating:** 6
**Confidence:** 3

**Summary:**

A novel method is proposed to derive the mode connectivity over the simplex defined by the central server for an improved global model as well as local personalization performances in the federated settings.

**Strengths:**

The proposed objective using Riesz s-Energy regularization along with the Euclidean projection onto the simplex is novel and convincing. The paper is well structured and various analyses are provided with accessible illustrations. The hyperparameter details provided are helpful in reproducing the results.

**Weaknesses:**

- Line 25: is leverages -> leverages
- Line 26, 33, 66, 129: weight parameter -> parameter (since parameters typically represents both weights and biases in NNs)
- Line 60: over the clients -> for the $k$-th client
- Line 96: (not sure) in the RHS, it may be $\boldsymbol{w}_\boldsymbol{\alpha}$
- Line 127: please consider adding equation numbers in each line of the pseudocode of Algorithm 1.
- Line 148: missing citation
- Lines 189-190: the baseline methods are not convincing. Since the proposed method explicitly induces the mode connectivity (i.e., line 25 per se) for the improved FL performance, it should have been compared to methods with similar motivations, e.g. [1], which was cited in the Related Works section (line 286) but not directly compared. Plus, authors should have also considered adding more personalization methods (e.g., [2], which shares the same strategy, i.e., only exchanging parameters of classification layer, for communication-efficient personalization in FL) as the authors aim to prove the effectiveness of `FLOCO` for improved personalization performances.
- Line 212: please also report standard deviations on 5 different runs.
- Lines 461-466: the communication rounds (100 and 500), the number of clients (100), and the number of per-round clients in CIFAR-10 (which is 30 out of 100) are somewhat deviated from the practical FL settings, which raises doubts about the scalability of the proposed method. It is more acceptable if the proposed method is specifically designed for the cross-silo FL settings, which usually assume a small number of stateful clients and (optionally) full participation across rounds.


[1] Connecting low-loss subspace for personalized federated learning (2022)
[2] Think Locally, Act Globally: Federated Learning with Local and Global Representations (2020)

**Questions:**

- Line 255: in Figure 2, the convergence speed of the global test accuracy & the average local test accuracy of the proposed method is slower than the baseline methods (when seeing round ~50), which is somewhat counterintuitive provided that the total gradient variance remains the smallest across all FL training rounds (which authors stated that it leads to the better performance, i.e., line 265-266). Could the authors please provide an explanation on this?
- Line 255: same in Figure 2 (and similarly in Figure 3), the average local test accuracy is suddenly surged, starting from the 200th round. This is quite intriguing, and I also expect an explanation on this result.

**Limitations:**

The authors do not evaluate the proposed method out of vision classification task.

---

> ### Author Rebuttal · Authors · 2024-08-07
>
> We would like to express our gratitude to Reviewer 44Bh for their detailed review. We are happy to read that the reviewer acknowledges the novelty of our method that exploits recent findings in mode connectivity to train a simplex in the FL setting, which improves local and global performance. Moreover, we appreciate that the reviewer found our proposed Riesz-energy-based Euclidean projection onto the unit simplex novel and convincing. Below, we will answer each of the reviewer’s weakness remark, question and limitation.
>
> ***Reviewer’s remark:*** Reviewer points out typos, missing citations, naming conventions, e.g. weight parameter -> parameter, and style changes, e.g. inclusion of equations numbers.
> ***Author’s answer:*** We have fixed all pointed out remarks above.
>
> ***Reviewer’s remark:*** Lines 189-190: the baseline methods are not convincing. Since the proposed method explicitly induces the mode connectivity (i.e., line 25 per se) for the improved FL performance, it should have been compared to methods with similar motivations, e.g. [1], which was cited in the Related Works section (line 286) but not directly compared. Plus, authors should have also considered adding more personalization methods (e.g., [2], which shares the same strategy, i.e., only exchanging parameters of classification layer, for communication-efficient personalization in FL) as the authors aim to prove the effectiveness of FLOCO for improved personalization performances.
> ***Author’s answer:*** We agree that the originally chosen baselines are not sufficient, and thus we report on new comparisons with 5 other global AND local FL baselines.  As shown in Figures 1 and 2 in the rebuttal pdf and Tables 1-4 in the global response, our FLOCO method outperforms all the tested baselines (the results were averaged over 5 different random seeds).  We implemented our FLOCO in the FL Bench [1] framework, and compared it with most of the baselines except SuPerFed, another state-of-the-art FL method that is not supported by FL Bench.  SuPerFed is implemented in our Flower-based framework, and compared with our original implementation of FLOCO.  Note that FLOCO run on the FL Bench framework and that run on our Flower-based framework do not exactly match because of the mismatch in the aggregation order of clients.  Due to the time limitation, we could have run experiments in the FL base framework only on CIFAR10 Dir(0.5) on a SimpleCNN.   For the camera-ready we will run these comparisons on all dataset split-model combinations.
>
> ***Reviewer’s remark:*** Line 212: please also report standard deviations on 5 different runs.
> ***Author’s answer:*** In Table 1-4 in the global response, we report on standard deviations over the 5 different seeded runs for each experiment.
>
> ***Reviewer’s remark:*** Lines 461-466: the communication rounds (100 and 500), the number of clients (100), and the number of per-round clients in CIFAR-10 (which is 30 out of 100) are somewhat deviated from the practical FL settings, which raises doubts about the scalability of the proposed method. It is more acceptable if the proposed method is specifically designed for the cross-silo FL settings, which usually assume a small number of stateful clients and (optionally) full participation across rounds.
> ***Author’s answer:*** We have tested our method on the FEMNIST dataset, a practical FL data set where different clients have different numbers of data points.
>
> ***Reviewer’s question 1:*** Line 255: in Figure 2, the convergence speed of the global test accuracy & the average local test accuracy of the proposed method is slower than the baseline methods (when seeing round ~50), which is somewhat counterintuitive provided that the total gradient variance remains the smallest across all FL training rounds (which authors stated that it leads to the better performance, i.e., line 265-266). Could the authors please provide an explanation on this?
> ***Author’s answer:*** The slow convergence of FLOCO in the beginning comes from the larger degrees of freedom (we train M end-points thus have M times more parameters to be trained). After the 200th round where we project the clients onto the solution simplex and thus the variance of gradient is significantly reduced, the global training is accelerated, as seen in Fig.2 (left) in the original submission (not as drastic as the local test accuracy, but still a larger slope than FedAve and DITTO can be observed).
>
> ***Reviewer’s question 2:*** Line 255: same in Figure 2 (and similarly in Figure 3), the average local test accuracy is suddenly surged, starting from the 200th round. This is quite intriguing, and I also expect an explanation on this result.
> ***Author’s answer:***  As mentioned in the caption of Fig. 2 in the submission, FLOCO performs the simplex learning (with alpha drawn from uniform distribution over the whole standard simplex) until the 200th round.  At the 200th round, FLOCO performs the proposed projection of the clients onto the simplex, based on the update signals, and starts using the solution simplex for personalization.  We will make this point clearer in the main text.
>
> ***Reviewer’s limitation remark:*** The authors do not evaluate the proposed method out of vision classification task.
> ***Author’s answer:*** We will extend our experimental setup to the google speech dataset for the camera-ready version.
>
> [1] Connecting low-loss subspace for personalized federated learning (2022)
> [2] Think Locally, Act Globally: Federated Learning with Local and Global Representations (2020)
> [3] FL Bench: (https://github.com/KarhouTam/FL-bench)

---

> ### Comment · Reviewer_44Bh · 2024-08-11
> **Reply to authors**
>
> Thanks for the authors' time and effort for my suggestions and questions.
> Especially, I really appreciate additional experiments that authors have committed to improve their initial manuscript.
>
> I have only one remaining concern that needs to be clarified further.
> > The question about the practicality of the current experimental setting & the authors' answer about the FEMNIST baseline being used.
>
> While it is acceptable that FEMNIST dataset[1] if $\texttt{Floco}$ is designed for real-world _cross-device_ federated setting, it seems that the authors did not use the full dataset as stated in Appendix B, line 464.
> In other words, the total number of clients in FEMNIST dataset is originally 3,597, but only 100 clients are sampled and used in this work.
> That is why I recommended authors to tone down and specify the scope of this work into __cross-silo__ FL setting, rather than sticking to an algorithm for generic FL setting.
>
> This can be further justified due to the design choice of the proposed method:
> * The $\mathcal{R}_k$ should be __kept and tracked__ by the server for each client $k\in[K]$ till the end of the federation round.
> * This means that the algorithm requires __stateful__ clients, which does not hold in the practical _cross-device_ setting. (please see Table 1 of [3])
> * Moreover, this statefulness inevitably forces the algorithm to have a moderate number of clients, $K$, which is consistent with the current choice of experimental settings.
>
> If I have misunerstood and if the proposed method is even scalable to the empirical settings with massive number of clients (e.g., original FEMNIST setting with $K=3,597$, StackOverflow dataset[2] with $K=342,477$, and iNaturalist dataset[4] with $K=9,275$), please enlighten me.
>
> [1] (2019) LEAF: A Benchmark for Federated Settings
> [2] (2019) https://www.tensorflow.org/federated/api_docs/python/tff/simulation/datasets/stackoverflow/load_data
> [3] (2019) Advances and Open Problems in Federated Learning
> [4] (2020) Federated Visual Classification with Real-World Data Distribution

---

> > ### Comment · Reviewer_a16f · 2024-08-11
> >
> > I second Rev 44Bh comments. As mentioned in my response, I also think the authors should comment on the scalability of their algorithm and the applicability on general cross-device scenarios. Looking forward to the authors response.

---

> > > ### Author Response · Authors · 2024-08-13
> > > **Practicality**
> > >
> > > Dear Reviewer 44Bh,
> > >
> > > Thank you for your additional comments.  We agree with the reviewer that we only tested our method on the sampled 100 clients in FEMNIST.  As the reviewer pointed out, our method, in particular in the subregion assignment, requires the statefulness of the clients. In the generic FL setting where many clients with few data samples exist, the projection and the subregion assignment in FLOCO are not reliable anymore.  Therefore, we expect that the low data clients are assigned random subregions, which will deteriorate the performance.  Following the suggestion by Reviewer 44Bh, as well as by Reviewer a16f, we revise the paper so that our target is not the generic FL setting, but the cross-silo FL setting, where each client observes sufficiently many data samples.  We appreciate the reviewer’s constructive suggestions.
> > >
> > > Best,
> > > Authors

---

> ### Comment · Reviewer_44Bh · 2024-08-13
> **Reply to authors**
>
> I sincerely appreciate the authors' positive answers.
> Someone might think that narrowing the scope of the work limits the scalability of one's proposed method, but I humbly believe that clarifying what can and can't be done in one's work adds more value to the research community as well as clarifies its practical implications.
> Thus, please reflect what the authors have agreed upon during discussion periods in their final manuscript.
> In conclusion, I decide to raise my score, and here are reasons.
>
> ### Why not a lower score
> * The authors made enough efforts in rebuttals to complement lacking and outdated baselines in their initial manuscript, and the results are convincing.
> * After the discussion, the authors decide to adjust the main scope of their work in a more acceptable direction.
> * The observation of the mitigating gradient variance in the proposed method (i.e., Fig. 2) is also intriguing and noteworthy.
>
> ### Why not a higher score
> * The scope (and possibly the title) of the manuscript should be refined according to the authors' acceptance of my suggestions (i.e., specifying the scope of the algorithm in a cross-silo FL setting), which will require some changes (e.g., re-writing of overall contents) to the status quo.
> * While I recognize and accept that the mode connectivity --- a main motivation and an ingredient of the proposed method --- is fairly an empirical observation, a theoretical or in-depth empirical analysis (e.g., generalization guarantee, or convergence analysis) that incorporates the mode connectivity for federated settings is missing. The authors could complement this perspective by building on the recent analysis. (e.g., [1,2])
> * (minor) Overall, the notations are difficult to parse and look similar to each other at first glance.
>
> [1] Proving Linear Mode Connectivity of Neural Networks via Optimal Transport (AISTATS'24)
> [2] Linear Connectivity Reveals Generalization Strategies (ICLR'23)

---

### Official Review · Reviewer_bjGi · 2024-07-13

**Soundness:** 3
**Presentation:** 3
**Contribution:** 2
**Rating:** 3
**Confidence:** 4

**Summary:**

The authors propose federated learning over connected modes (FLOCO), where clients are assigned local subregions in this simplex based on their gradient signals, and together learn the shared global solution simplex.

**Strengths:**

This paper is richer in the type of experiments.

**Weaknesses:**

The proposed methodology lacks innovation.

The experimental baseline is too old (2021) and there are many recent and similar clustered federated algorithms that lack comparison.

**Questions:**

1 Can the authors clarify the difference between the proposed method and other clustering federated learning algorithms?

2 Can you specifically describe the implementation details of the algorithm? Or improve the description. For example, there is a lack of more detailed description for the generation of R.

3 In the algorithm section, can the specific algorithm be explained more clearly?

**Limitations:**

The authors discuss the reasons for the poor experimental performance of the proposed methods

---

> ### Author Rebuttal · Authors · 2024-08-07
>
> We are deeply thankful for Reviewer bjGi for their thorough review. We are pleased to read that the reviewer appreciates the richness of our experiments, the soundness and presentation of our work. In the In the following, we answer the reviewer’s remarks and questions in more detail:
>
> ***Reviewer’s remark:*** “The proposed methodology lacks innovation.”
> ***Author’s answer:***   We argue that our paper has significant novelty and contributions: Our way of using solution simplexes for federated learning is unique; we employ a novel projection method that spreads FL clients well across the simplex;  and we show in many different experiments that FLOCO performs well for global AND personalized FL, which are usually problems tackled separately. Thus we would like to ask for clarification by the reviewer about why our method lacks innovation.
>
> ***Reviewer’s remark:*** “The experimental baseline is too old (2021) and there are many recent and similar clustered federated algorithms that lack comparison.”
> ***Author’s answer:*** We agree that the original baselines are old, and thus we report on new comparisons with 5 other global AND local FL baselines.  As shown in Figures 1 and 2 in the rebuttal pdf and Tables 1-4 in the global response, our FLOCO method outperforms all the tested baselines (the results were averaged over 5 different random seeds).  We implemented our FLOCO in the FL Bench [1] framework, and compared it with most of the baselines except SuPerFed, another state-of-the-art FL method that is not supported by FL Bench.  SuPerFed is implemented in our Flower-based framework, and compared with our original implementation of FLOCO.  Note that FLOCO run on the FL Bench framework and that run on our Flower-based framework do not exactly match because of the mismatch in the aggregation order of clients.  Due to the time limitation, we could have run experiments in the FL base framework only on CIFAR10 Dir(0.5) on a SimpleCNN.   For the camera-ready we will run these comparisons on all dataset split-model combinations.
>
> ***Reviewer’s question 1:*** Can the authors clarify the difference between the proposed method and other clustering federated learning algorithms?
> ***Author’s answer:***  Our method does not cluster anything but project clients onto the solution simplex, so that the degrees of freedom within the simplex is used to capture characteristics of each client.
>
> ***Reviewer’s question 2:*** Can you specifically describe the implementation details of the algorithm? Or improve the description. For example, there is a lack of more detailed description for the generation of R.
> ***Author’s answer:***
> R is simply a L1-ball around the projected point of each client.  For drawing samples from the uniform distribution over R, we perform the following:
> Generate random points on the surface of the L1 unit ball by normalizing the absolute values of a randomly generated vector.
> Scale these points to fit within the desired radius.
> Translate the points by the given center.
> We will add further algorithm details in Appendix.
>
> ***Reviewer’s question 3:*** In the algorithm section, can the specific algorithm be explained more clearly?
> ***Author’s answer:***  We will add further algorithm details in Appendix.  We are happy to clarify its algorithm details if the reviewer specifies on which algorithm our description is unclear, other than the generation of samples on R.
>
> ***Reviewer’s limitation remark:*** The authors discuss the reasons for the poor experimental performance of the proposed methods
> ***Author’s answer:***  We do not understand this comment.  We do not show poor performance of the proposed method, and thus do not discuss the reason.  Could you please elaborate more?
>
> [1] FL Bench: (https://github.com/KarhouTam/FL-bench)

---

> ### Author Response · Authors · 2024-08-13
> **Response?**
>
> Dear Reviewer bjGi,
>
> Since the discussion deadline is approaching, we would kindly ask Reviewer bjGi to respond to our rebuttal comments.  We see that our rebuttal comments addressed most of the criticisms by the reviewer, and would like to know the reasons why the reviewer still keeps the clear rejection score 3 after our rebuttal, which is a clear outlier among the other reviewers’ scores.  We look forward to your response.
>
> Best,
> Authors

---

> ### Comment · Reviewer_bjGi · 2024-08-13
>
> Thank you for the rebuttal. After reading the rebuttal and reviews of other reviewers, I still maintain that this is a borderline paper. I will maintain my original rate.

---

> > ### Author Response · Authors · 2024-08-13
> >
> > Dear Reviewer,
> >
> > Thank you very much for your response. Since the reviewer said that our submission is a borderline paper and the reviewer would keep the original score "3: Reject", we wonder if the reviewer perhaps misunderstands the scoring system of NeurIPS. The NeurIPS review system adopts the scoring system NOT 1-5 but 1-10, and the score "3: Reject" that the reviewer gave to our paper is not for the borderline papers, as the reviewer assessed our paper in the previous response, but for the papers that should be rejected. Therefore, we would kindly ask the reviewer to adjust the score, keeping in mind the scoring system that the NeurIPS committee has defined, i.e., 2: Strong reject, 3: Reject, 4: Borderline Reject, 5: Borderline Accept, 6: Weak Accept.
> >
> > Additionally, if the reviewer still leans toward a reject, giving, say score 4: Borderline Reject, we would kindly ask the reviewer to provide the reasons for rejection by explaining which of our responses is not sufficient, and more importantly, why.
> >
> > Kind regards,
> > Authors

---

### Author Rebuttal · Authors · 2024-08-07

We would like to thank all reviewers for their detailed review, including remarks, questions and suggestions for improvement. As most reviewer’s pointed out, some important baselines, such as SuPerFed[1] and others are crucial to benchmark against our method. We have thus extended our experimental setting and have included 5 more baselines. As our new results suggest, FLOCO outperforms all benchmarked methods including the state-of-the-art.
The main experimental results that we have computed during the rebuttal phase are listed in the 4 tables below. We reference them throughout our whole rebuttal.
Tables 1 & 2 show the benchmark of FLOCO and another method that employs mode connectivity which is called SuPerFed, for global and average local test performance respectively.
Tables 3 & 4 show the benchmark of FLOCO against 5 baselines from the FL-Bench repository, for global and local test performance respectively.
In the rebuttal pdf, in Figures 1 & 2 we have included the training curves for the latter. Moreover, we have included a comparison plot of FedAvg with simplex learning compared to Floco, as suggested by Reviewer a16f in Figures 3 & 4.
Lastly, Figures 5 & 6 in the rebuttal pdf show how Floco performs when training a simplex over each parameter in each layer. As expected, due to the higher degrees of freedom, Floco over all parameters, Floco-All, needs more time to converge to the same performance as Floco with simplex learning only on the last layer.


[1] Connecting low-loss subspace for personalized federated learning (2022)

[2] FL Bench: (https://github.com/KarhouTam/FL-bench)

# Table 1: Floco against SuPerFed - Global test accuracy:

| Model-Dataset-Split                    | FLOCO | SuPerFed |
|----------------------------------------|-------|----------|
| SimpleCNN - CIFAR10 - 5-Fold           | 68.26 | 63.22    |
| SimpleCNN - CIFAR10 - Dir(0.5)         | 69.79 | 63.00    |
| SimpleCNN - FEMNIST - -                | 77.95 | 76.80    |
| PretrainedResnet18 - CIFAR10 - 5-Fold  | 74.61 | 64.88    |
| PretrainedResnet18 - CIFAR10 - Dir(0.5)| 79.11 | 76.04    |

# Table 2: Floco against SuPerFed - Avg. local test accuracy:

| Model-Dataset-Split                    | FLOCO | SuPerFed |
|----------------------------------------|-------|----------|
| SimpleCNN - CIFAR10 - 5-Fold           | 80.92 | 75.65    |
| SimpleCNN - CIFAR10 - Dir(0.5)         | 74.64 | 71.73    |
| SimpleCNN - FEMNIST - -                | 83.89 | 82.59    |
| PretrainedResnet18 - CIFAR10 - 5-Fold  | 87.38 | 52.78    |
| PretrainedResnet18 - CIFAR10 - Dir(0.5)| 82.29 | 60.91    |

# Table 3: Floco against 5 baselines - Global test accuracy

| Model-Dataset-Split                    | FLOCO | Ditto | FedAvg | FedRep | FedPac | FedDyn |
|----------------------------------------|-------|-------|--------|--------|--------|--------|
| SimpleCNN - CIFAR10 - Dir(0.5)         | 73.44 | 73.06 | 72.94  | 43.70  | 68.70  | 31.67  |

# Table 4: Floco against 5 baselines - Avg. local test accuracy

| Model-Dataset-Split                    | FLOCO | Ditto | FedAvg | FedRep | FedPac | FedDyn |
|----------------------------------------|-------|-------|--------|--------|--------|--------|
| SimpleCNN - CIFAR10 - Dir(0.5)         | 79.54 | 79.20 | 73.42  | 68.00  | 73.15  | 31.39  |






# Extended answer to reviewer a16f:
***Reviewer’s question 7:*** Basically, the method only applies to the last FC layer - did the authors try it on the full model?
***Author’s answer:*** We show the result with FLOCO-all in Figures 5 and 6 in the rebuttal pdf, where we observe that applying FLOCO to all layers significantly reduces training speed.

---

### Comment · Area_Chair_9U76 · 2024-08-11
**Start discussions with authors, please**

Dear Reviewers,

Thank you for your valuable contributions to the review process. As we enter the discussion phase (August 7-13), I kindly request your active participation in addressing the authors' rebuttals and engaging in constructive dialogue.

Please:

- Carefully read the authors' global rebuttal and individual responses to each review.

- Respond to specific questions or points raised by the authors, especially those requiring further clarification from you.

- Engage in open discussions about the paper's strengths, weaknesses, and potential improvements.

- Be prompt in your responses to facilitate a meaningful exchange within the given timeframe.

- Maintain objectivity and professionalism in all communications.

If you have any concerns or need guidance during this process, please don't hesitate to reach out to me.

Your continued engagement is crucial for ensuring a fair and thorough evaluation process.
Thank you for your dedication to maintaining the high standards of NeurIPS.

Best regards,

Area Chair

---

### Decision · Program_Chairs · 2024-09-25

**Decision:**

Accept (poster)

**Comment:**

**Summary:**

FLOCO proposes using linear mode connectivity to identify a solution simplex in neural network weight space for federated learning. Clients are assigned subregions in the simplex based on gradient signals to allow personalization while jointly learning a global solution. Claims to accelerate global training and improve local accuracy compared to baselines.

**Strengths:**

- Novel application of linear mode connectivity/simplex learning to federated learning
- Well-written paper with comprehensive experiments
- Addresses both global and personalized FL in a unified framework
- Achieves good performance on both global and local accuracy metrics

**Weaknesses:**

- Limited theoretical analysis
- Some baseline comparisons missing initially (addressed in rebuttal)
- Scalability to very large numbers of clients unclear
- Applicability may be limited to cross-silo rather than cross-device FL settings

**Key Discussion Points:**

- *Baseline comparisons:* Authors added several new baselines in rebuttal, showing FLOCO outperforms them
- *Applicability:* Authors acknowledged FLOCO is more suited for cross-silo FL with stateful clients, not cross-device scenarios
- *Scalability:* Questions raised about performance with thousands of clients; authors agreed this is a limitation for low-data clients
- *Ablation studies:* Additional experiments requested and provided to isolate effects of different components
- *Theoretical analysis:* Lack of convergence analysis noted; authors agreed this is future work
- *Implementation details:* Clarifications provided on simplex learning process

Overall, reviewers generally found the core idea interesting and results promising, but had some concerns about scope/applicability and analysis depth. Scores ranged from borderline reject to weak accept, with most leaning positive after author responses. The consensus seems to be that this is a solid contribution to federated learning research, particularly for cross-silo scenarios, though some limitations remain.